# GeMS: Efficient Gaussian Splatting for Extreme Motion Blur

## Abstract

We introduce *GeMS*, a framework for 3D Gaussian Splatting designed to handle severely motion-blurred images. State-of-the-art deblurring method for extreme motion blur, such as ExBluRF, as well as Gaussian Splatting-based approaches like Deblur-GS, typically assume access to corresponding sharp images for camera pose estimation and point cloud generation, which is an unrealistic assumption. Additionally, methods relying on COLMAP initialization, such as BAD-Gaussians, fail due to the lack of reliable feature correspondences in cases of severe motion blur. To address these challenges, we propose *GeMS*, a 3D Gaussian Splatting (3DGS) framework that reconstructs scenes directly from extremely motion-blurred images. GeMS integrates: (1) *VGGSfM*, a deep learning-based Structure from Motion (SfM) pipeline which estimates camera poses and generates point clouds directly from severely motion-blurred images; (2) *3DGS-MCMC (Markov Chain Monte Carlo)* enables robust scene initialization by treating Gaussians as samples from an underlying probability distribution, eliminating heuristic densification and pruning strategies; and (3) Joint optimization of camera motion trajectory and Gaussian parameters which ensures stable and accurate reconstruction. While this pipeline produces reasonable reconstructions, extreme motion blur can still introduce inaccuracies, especially when all input views are severely blurred. To address this, we propose *GeMS-E*, which integrates a progressive refinement step when event data is available. Specifically, we perform (4) *Event-based Double Integral (EDI)* deblurring, which first restores deblurred images from motion-blurred inputs using events. These deblurred images are then fed into the GeMS framework, leading to improved pose estimation, point cloud generation, and hence overall reconstruction quality. Both GeMS & GeMS-E achieve state-of-the-art performance on synthetic as well as real-world datasets, demonstrating their effectiveness in handling extreme motion blur. To the best of our knowledge, we are the first to effectively address this motion deblurring problem in extreme blur scenarios within a 3D Gaussian Splatting framework directly from severely motion blurred images.

## 1 Introduction

Motion blur is a fundamental and unsolved challenge for *3D scene reconstruction* and *novel view synthesis*, especially in real-world scenarios involving *high-speed camera motion* or *low-light conditions*. Such conditions are ubiquitous in practical settings ranging from robotics and autonomous vehicles to handheld photography, where capturing sharp images is often impossible. Robust 3D reconstruction from motion-blurred inputs is therefore essential for advancing computer vision systems.

Despite recent advances such as Neural Radiance Fields (NeRF) Mildenhall et al. (2020) and 3D Gaussian Splatting (3DGS) Kerbl et al. (2023b), these methods fundamentally rely on sharp images and accurate camera poses. Traditional Structure-from-Motion (SfM) pipelines like COLMAP Schonberger & Frahm (2016) are especially brittle under heavy blur: they depend on reliable feature correspondences for keypoint detection and matching, but blur severely degrades texture information, leading to unreliable matches, poor pose estimation, and ultimately, failure to reconstruct the scene. As a result, there is currently no practical solution for robust 3D reconstruction from extremely motion-blurred images without sharp-image supervision, a critical gap in the literature.

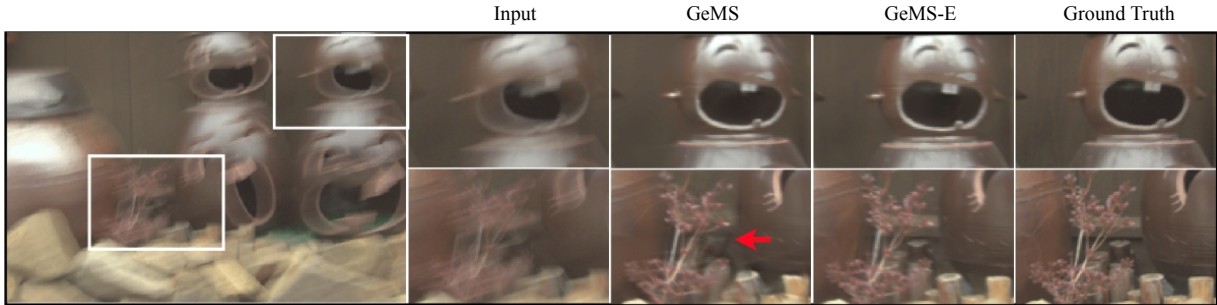

**Figure 1: Overview:** GeMS reconstructs sharp 3D scenes directly from extremely motion-blurred images without relying on COLMAP or sharp image supervision. GeMS-E leverages event streams, when available, to further enhance reconstruction quality in the most challenging cases.

To address this open problem, we introduce **GeMS**, a fundamentally new, efficient 3D Gaussian Splatting framework that reconstructs sharp 3D scenes directly from severely blurred inputs. Our approach is not a simple stacking of existing techniques, but a tightly integrated, self-correcting system built on three core innovations:

**Blur-Robust Initialization with VGGSfM:** We replace COLMAP's brittle feature matching with VG-GSfM Wang et al. (2024), a deep, differentiable SfM framework. VGGSfM's learned 2D point tracking and end-to-end optimization provide robust, blur-tolerant initialization of camera poses and point clouds, succeeding where all classical and many recent methods fail. This enables downstream modules to operate even in the presence of extreme motion blur.

**Probabilistic Scene Modeling with 3DGS-MCMC:** We incorporate a probabilistic formulation using 3DGS-MCMC Kheradmand et al. (2025), which treats Gaussians as samples from a scene distribution and uses Markov Chain Monte Carlo (MCMC) sampling to adaptively densify and refine geometry. This avoids the heuristic and brittle densification strategies of prior work, maintaining high reconstruction quality even when the input is sparse or noisy due to extreme motion blur.

**Joint Trajectory-Geometry Optimization:** Because motion blur arises from continuous camera motion during exposure, we jointly optimize camera trajectories (using Bézier curves) and Gaussian parameters with physics-based losses Zhao et al. (2024). This joint optimization is essential for aligning the reconstructed geometry with the actual blur formation process and correcting errors that would otherwise propagate through the pipeline.

The synergy of these components is critical: VGGSfM's robust initialization enables effective probabilistic refinement, while joint optimization ensures global consistency and error correction. Our systematic experiments across a spectrum of blur levels demonstrate that each module is necessary, and their integration yields state-of-the-art performance. GeMS consistently outperforms both traditional and recent methods, especially as blur severity increases.

For extreme blur cases where all input views are severely blurred, we further extend our framework to **GeMS-E**, when event data is available. By incorporating event-based deblurring using the Event-based Double Integral (EDI) model Pan et al. (2019), GeMS-E leverages high-temporal-resolution event data to recover sharp images. These EDI deblurred images are seamlessly fed to our GeMs framework. This extension outperforms all event-driven baselines in both synthetic and real-world datasets.

Comprehensive experiments demonstrate that GeMS and GeMS-E achieve state-of-the-art results, outperforming existing methods in both synthetic and real-world motion-blur scenarios. Our GeMs/GeMS-E results are illustrated in Figure 1

Our key contributions are as follows:

- We propose GeMS, an efficient 3D Gaussian Splatting framework that enhances robustness in extreme motion-blurred scenarios by leveraging VGGSfM for camera pose estimation and point cloud

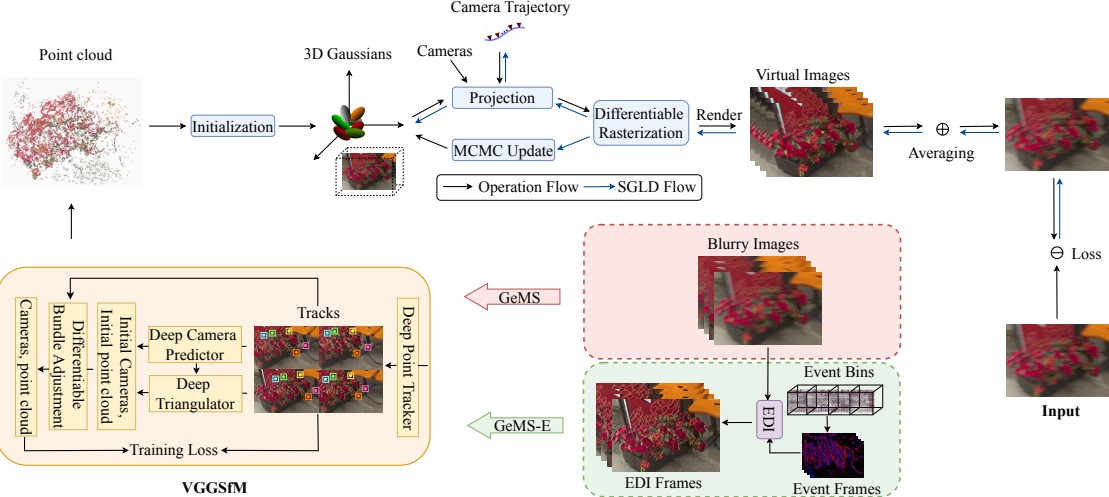

**Figure 2: Our Method:** *GeMS* addresses extreme motion blur in 3D Gaussian Splatting framework. GeMS directly optimizes Gaussians on blurred images without requiring COLMAP or sharp supervision, leveraging VGGSfM for robust SfM initialization, 3DGS-MCMC with joint optimization for effective refinement of camera poses and Gaussian parameters. When event data is available, GeMS-E enhances this by incorporating the EDI model to recover deblurred images, which are then fed into the GeMS pipeline for improved reconstructions.

generation without sharp image supervision. The framework further incorporates MCMC-based Gaussian initialization and optimization, along with joint optimization of camera poses and Gaussian parameters, to effectively restore sharp scene representations.

- When event data is available, GeMS-E integrates event-based deblurring using the EDI model, enabling sharp image reconstruction and novel sharp view synthesis even in extreme motion blur conditions, particularly in cases where all input views are severely blurred. To further support research in this area, we introduce a complementary synthetic event dataset *EveGeMS* specifically curated for extreme blur scenarios.

- We conduct a systematic analysis across multiple blur levels, demonstrating the robustness of both VGGSfM and 3DGS-MCMC modules under severe motion blur.

- Through extensive evaluations on both synthetic and real-world datasets, we demonstrate superior performance in deblurring and novel view synthesis, achieving state-of-the-art reconstruction quality while significantly improving computational and memory efficiency.

To the best of our knowledge, this is the first work to effectively address the challenge of reconstructing sharp 3D scene from extremely motion blurred images within a 3D Gaussian Splatting framework, without relying on impractical sharp image supervision for pose and point cloud initialization.

## 2 Method

*GeMS* is an efficient 3D Gaussian Splatting framework for deblurring and sharp novel view synthesis under extreme motion blur. It leverages VGGSfM Wang et al. (2024) for camera pose estimation and point cloud generation, followed by MCMC-based Gaussian initialization and optimization Kheradmand et al. (2025). A joint optimization of camera trajectories and Gaussian parameters further refines the reconstruction, ensuring robustness in highly blurred scenarios. GeMS-E (Event-Assisted Deblurring): When event data is available, we first recover sharp images using the EDI model Pan et al. (2019). These deblurred images are then processed through our GeMS framework to facilitate both motion deblurring and sharp novel view synthesis, even in the presence of extreme motion blur across all input images. Our overall framework is illustrated in Figure 2. Background of 3D Gaussian Splatting is presented in appendix B.1.

## 2.1 GeMS (Direct Deblurring)

COLMAP Schonberger & Frahm (2016) is the de facto choice for structure-from-motion (SfM) initialization in NeRF and 3D Gaussian Splatting (3DGS) pipelines. However, it requires sharp, high-quality images for reliable feature matching, rendering it ineffective under severe motion blur. Consequently, state-of-the-art deblurring methods that rely on COLMAP become ineffective in such challenging scenarios. In contrast, our proposed GeMS framework eliminates dependence on COLMAP by employing a robust and efficient Gaussian Splatting pipeline that integrates VGGSfM with 3DGS-MCMC based optimization. We then jointly optimize Gaussian parameters and camera motion trajectories to refine pose inaccuracies. This systematic approach enables accurate and sharp 3D scene reconstruction directly from severely motion-blurred images.

### 2.1.1 Robust Initialization and Optimization with VGGSfM and 3DGS-MCMC

Accurate initialization is critical for 3D Gaussian Splatting (3DGS), especially in scenarios with severe motion blur where traditional methods like COLMAP fail due to unreliable feature correspondences. This limitation affects several state-of-the-art deblurring pipelines such as ExBluRF Lee et al. (2023) and BAD-Gaussians Zhao et al. (2024), which depend on COLMAP initialization. To overcome this, we integrate two complementary techniques: VGGSfM, a fully differentiable deep-learning-based Structure-from-Motion (SfM) pipeline for robust camera pose estimation and point cloud generation; and MCMC-based Gaussian Splatting with joint optimization, which treats Gaussians as probabilistic samples and jointly optimizes camera poses and Gaussian parameters for adaptive and accurate scene reconstruction.

**VGGSfM for SfM initialization:** Unlike traditional SfM pipelines that rely on incremental image registration and brittle feature matching, VGGSfM estimates all camera poses in an end-to-end manner. It leverages recent advances in deep 2D point tracking to extract reliable, pixel-accurate tracks without explicit pairwise feature matching, ensuring robustness even when image textures are severely degraded by blur. Rather than gradual registration, VGGSfM employs a Transformer-based model for global pose estimation, significantly improving stability under motion blur. Furthermore, it integrates a differentiable bundle adjustment layer based on the Theseus solver, enabling joint optimization of camera poses and 3D points within a learning framework. This architecture allows VGGSfM to produce valid initializations at extreme blur levels where COLMAP fails. We presented systematic evaluations to validate the effectiveness of VGGSfM at various blur levels in Section 3.4.1.

**3DGS-MCMC for joint optimization:** Building on the 3DGS-MCMC framework Kheradmand et al. (2025), we reinterpret Gaussian splatting as a probabilistic sampling process. Gaussians are treated as samples drawn from an underlying scene distribution and updated using Stochastic Gradient Langevin Dynamics (SGLD). This probabilistic approach allows Gaussians to dynamically relocate to high-likelihood regions, eliminating the need for heuristic cloning or pruning. Systematic evaluations with resepct to various blur levels are presented in Section 3.4.1.

To address inaccuracies in camera poses introduced by severe motion blur, we incorporate joint optimization of Bézier motion trajectories and Gaussian parameters. The joint optimization is essential because motion blur fundamentally arises from continuous camera motion during exposure. Initial poses from VGGSfM provide only discrete viewpoints, failing to capture the true motion path responsible for observed blur. By jointly optimizing trajectories and geometry, we directly model the physics of motion blur, where each blurry image integrates sharp scene content along the camera's continuous trajectory. This co-adaptation enables iterative correction of both trajectory errors and geometry misalignments, ensuring the synthesized blur matches real input blur and significantly reducing artifacts. By modeling and optimizing the camera trajectory alongside scene geometry, we ensure geometric and photometric consistency and improve reconstruction accuracy.

**Synergistic integration of VGGSfM and 3DGS-MCMC with joint optimization:** The integration of VGGSfM and 3DGS-MCMC with joint optimization creates a truly synergistic, end-to-end differentiable pipeline that goes far beyond simply stacking existing techniques. After initialization with VGGSfM, the probabilistic scene modeling of 3DGS-MCMC and the joint optimization of camera poses and scene geometry are tightly coupled, allowing gradients and error signals to flow seamlessly across modules during

training. This design enables each part of the system to actively compensate for the limitations of the others: 3DGS-MCMC adaptively refines and densifies the often noisy or sparse outputs from VGGSfM, while joint optimization co-refines Bézier camera trajectories and Gaussian scene parameters to ensure physical and photometric consistency with the observed data.

### 2.1.2 Physical Motion Blur Image Formation Model

The process of image formation in a digital camera involves the accumulation of photons during the exposure period, which are subsequently converted into electrical signals. Mathematically, this can be expressed as an integration over a sequence of virtual latent sharp images:

$$\mathbf{B}(\mathbf{u}) = \phi \int_0^\tau \mathbf{C}_t(\mathbf{u}) \mathrm{dt} \tag{1}$$

where $\mathbf{B}(\mathbf{u}) \in \mathbb{R}^{\mathrm{H} \times \mathrm{W} \times 3}$ represents the captured motion-blurred image, with $\mathbf{u} \in \mathbb{R}^2$ denoting the pixel location in an image of height H and width W. Here, $\phi$ is a normalization factor, $\tau$ is the exposure time, and $\mathbf{C}_t(\mathbf{u}) \in \mathbb{R}^{\mathrm{H} \times \mathrm{W} \times 3}$ corresponds to the latent sharp image at a given timestamp $t \in [0, \tau]$. The motion-blurred image $\mathbf{B}(\mathbf{u})$ arises due to camera movement during the exposure and is effectively the average of all latent sharp images $\mathbf{C}_t(\mathbf{u})$ over time. In practice, this integral is approximated using a finite number $n$ of discrete samples, leading to the following discrete formulation:

$$\mathbf{B}(\mathbf{u}) \approx \frac{1}{n} \sum_{i=0}^{n-1} \mathbf{C}_i(\mathbf{u}) \tag{2}$$

The extent of motion blur in an image is influenced by the camera's movement during exposure. A rapidly moving camera within a given exposure time creates severe motion blur. Note that $\mathbf{B}(\mathbf{u})$ remains differentiable with respect to each virtual sharp image $\mathbf{C}_i(\mathbf{u})$, which is a key property for optimization in motion deblurring tasks.

### 2.1.3 Camera Motion Trajectory Modeling in 3DGS-MCMC for Pose optimization

To model camera motion during exposure, we parameterize the pose in Special Euclidean group in 3 dimensions ($SE(3)$), which is a mathematical structure that describes all possible rigid body transformations in 3D space. While BAD-Gaussians Zhao et al. (2024) employ *linear* and *cubic spline* interpolation for trajectory estimation, these methods prove inadequate for handling severe motion blur. Hence our method adopts *Bézier curve interpolation* inspired from Chen & Liu (2024), which provides a smoother and more accurate representation. Given a Bézier curve of degree $M$, the camera motion is represented using $M + 1$ control points $T_j$ $(j = 0, ..., M)$. The interpolated camera pose at time $t$ is given as:

$$T_t = \prod_{j=0}^M \exp\left( \binom{M}{j} (1-u)^{M-j} u^j \cdot \log(T_j) \right) \tag{3}$$

where $u = t/\tau \in [0, 1]$ and $\tau$ is the exposure time. This formulation ensures smooth motion trajectory estimation while remaining differentiable, enabling joint optimization for accurate deblurring.

### 2.1.4 Loss Functions

Given a set of $K$ motion-blurred images, the goal is to jointly estimate the camera motion trajectory for each image and the learnable parameters of 3DGS, $\boldsymbol{\theta}$ (i.e., mean position $\boldsymbol{\mu}$, 3D covariance $\boldsymbol{\Sigma}$, opacity $\mathbf{o}$, and color $\mathbf{c}$). For this joint estimation framework, we draw inspiration from BAD-Gaussians Zhao et al. (2024). This estimation is achieved by minimizing the following loss function, which combines an $\mathcal{L}_1$ loss with a D-SSIM (1-SSIM) term. The loss is computed between $\mathbf{B}_k(\mathbf{u})$, the $k^{\mathrm{th}}$ synthesized blurry image generated via 3DGS and its corresponding real captured counterpart, $\mathbf{B}_k^{gt}(\mathbf{u})$.

$$\mathcal{L} = (1 - \lambda)\mathcal{L}_1 + \lambda \mathcal{L}_{\text{D-SSIM}} \tag{4}$$

### 2.1.5 Joint Optimization

To optimize both the learnable Gaussian parameters $\boldsymbol{\theta}$ and the camera poses $\mathbf{T}$ (represented using a Bézier curve of degree $M$ with $M+1$ control points $T_j$, $j = 0, ..., M$), the required Jacobians are computed to ensure proper gradient flow. As shown in Zhao et al. (2024), the gradient of the loss with respect to $\boldsymbol{\theta}$ is given by:

$$\frac{\partial \mathcal{L}}{\partial \boldsymbol{\theta}} = \sum_{k=0}^{K-1} \frac{\partial \mathcal{L}}{\partial \mathbf{B}_k} \cdot \frac{1}{n} \sum_{i=0}^{n-1} \frac{\partial \mathbf{B}_k}{\partial \mathbf{C}_i} \frac{\partial \mathbf{C}_i}{\partial \boldsymbol{\theta}} \tag{5}$$

while the gradient with respect to the camera pose is:

$$\frac{\partial \mathcal{L}}{\partial \mathbf{T}} = \sum_{k=0}^{K-1} \frac{\partial \mathcal{L}}{\partial \mathbf{B}_k} \cdot \frac{1}{n} \sum_{i=0}^{n-1} \frac{\partial \mathbf{B}_k}{\partial \mathbf{C}_i} \frac{\partial \mathbf{C}_i}{\partial \boldsymbol{\theta}} \frac{\partial \boldsymbol{\theta}}{\partial \mathbf{T}} \tag{6}$$

For clarity, the explicit dependence on $\mathbf{u}$ in $\mathbf{B}_k(\mathbf{u})$ and $\mathbf{C}_i(\mathbf{u})$ is omitted. The camera poses are parameterized using their corresponding Lie algebra representations in $SE(3)$, each expressed as a 6D vector.

## 2.2 GeMS-E (Event-Assisted Deblurring)

When event data is available, we extend our pipeline with GeMS-E to tackle scenarios where all input views are severely motion-blurred. In this setting, we first use the event camera data to generate sharp deblurred images, which are then passed through our GeMS pipeline. Specifically, these deblurred images are used exclusively for initializing the SfM stage (VGGSfM), enabling accurate camera pose estimation and sparse geometry that would otherwise be unattainable from the blurred inputs alone. Importantly, during the subsequent reconstruction process, we do not use the deblurred images for supervision; instead, the optimization is guided by the original blurry images using a physics-based blur formation model. The photometric loss is computed with respect to the observed blurred inputs, ensuring that the final scene reconstruction remains consistent with the actual captured data. This selective integration allows GeMS-E to benefit from robust event-based initialization while maintaining physically faithful blur-aware optimization, resulting in superior performance under extreme motion blur conditions.

### 2.2.1 Event Generation

Unlike frame-based cameras that record pixel brightness at a fixed frame rate, event cameras asynchronously generate an event $\mathbf{e}(x, y, \tau, p)$ when the change in brightness of pixel $(x, y)$ in the logarithmic domain exceeds a threshold $\Theta$ at time $\tau$.

$$p_{x,y,\tau} = \begin{cases} -1, & \log(\mathcal{I}_{x,y,\tau}) - \log(\mathcal{I}_{x,y,\tau-\Delta\tau}) < -\Theta \\ +1, & \log(\mathcal{I}_{x,y,\tau}) - \log(\mathcal{I}_{x,y,\tau-\Delta\tau}) > \Theta \end{cases} \tag{7}$$

where $p$ denotes the direction of the brightness change, and $\mathcal{I}_{(x,y,\tau)}$ represents the brightness value of pixel $(x, y)$ at time $\tau$. Since events are generated asynchronously, they are typically grouped into $b$ event bins, divided equally over time, to facilitate processing. Given a blurred image with exposure time from $t_{\text{start}}$ to $t_{\text{end}}$ and the associated event data $\{\mathbf{e}_i\}_{t_{\text{start}} < \tau_i \leq t_{\text{end}}}$, we can generate $\{B_k\}_{k=1}^{b}$ as follows:

$$B_k = \{\mathbf{e}_i(x_i, y_i, \tau_i, p_i)\}_{t_{k-1} < \tau_i \leq t_k} \tag{8}$$

where $t_k = t_{\text{start}} + \frac{k}{b} t_{\text{exp}}$ is the time division point between bins and $t_{\text{exp}} = t_{\text{end}} - t_{\text{start}}$ represents the exposure time. The EDI deblurring method is detailed in appendix C.1.

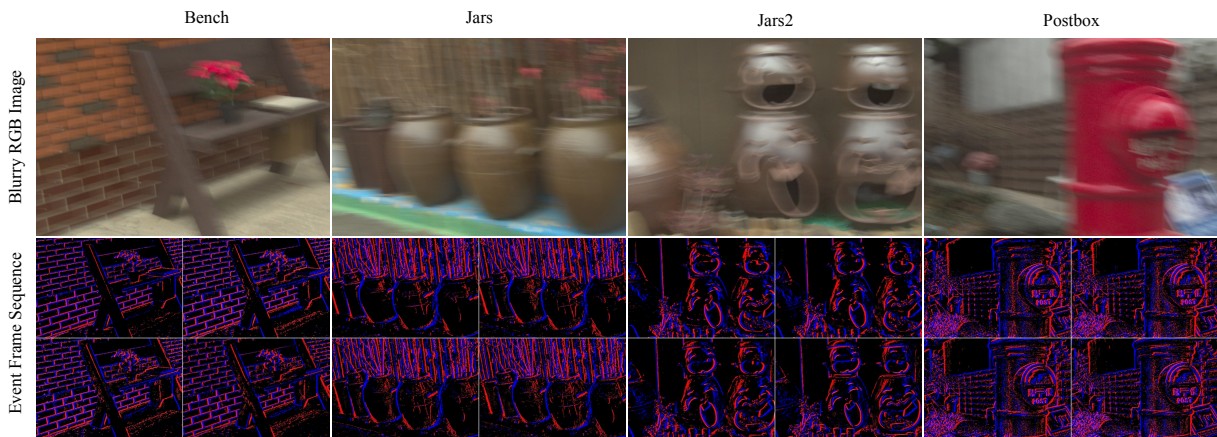

**Figure 3: Our Complementary Event Dataset (EveGeMS):** We present a synthetic event dataset designed for scenarios involving extreme motion blur. While the RGB frames suffer from severe blur, each is accompanied by a sequence of event frames that preserve fine structural and motion cues. This complementary information facilitates robust and accurate deblurring and novel view synthesis especially where all input views are severely blurred.

## 3 Experiments

### 3.1 Experimental Setup

**Datasets:** For evaluation, we use the synthetic dataset provided by ExBluRF Lee et al. (2023), which includes eight diverse outdoor scenes captured with challenging camera motion. Each scene contains 20 to 40 blurry views for training and 4 to 6 test views for evaluating novel view synthesis. Each blurry image in ExBluRF is paired with sequences of sharp images, which we utilize to create our complementary synthetic event dataset, *EveGeMS*, as shown in Figure 3. Specifically, the sharp frames recorded during the camera motion are processed using ESIM Rebecq et al. (2018) to generate the corresponding event stream, similar to the synthetic datasets in E2NeRF. Additionally, we use a real-world dataset from E2NeRF Qi et al. (2023), captured with the DAVIS346 color event camera. This dataset consists of five challenging scenes (i.e. letter, lego, camera, plant, and toys) with complex textures and varied motion, where RGB frames were captured with a 100ms exposure time, resulting in motion blur and complex camera trajectories. By incorporating event streams into the ExBluRF dataset, we contribute an extreme blurry and event pair dataset. We will release our complementary synthetic event dataset publicly upon acceptance to support future research in event-based deblurring and view synthesis under severe motion blur conditions. The complete event frame sequences of our EveGeMS dataset for all scenes are provided in the supplementary material.

**Baseline Methods and Evaluation Metrics:** Our baselines include: state-of-the-art deep learning-based single-image motion deblurring methods (MPRNetZamir et al. (2021), RestormerZamir et al. (2022)); event-based motion deblurring methods (EDIPan et al. (2019), E2NeRFQi et al. (2023), EBADNeRFQi et al. (2024)); and motion deblurring method designed for extreme motion blur (ExBluRF*Lee et al. (2023)) and 3D Gaussian Splatting-based deblurring (BAD-Gaussians* Zhao et al. (2024)). Note that both ExBluRF* and BAD-Gaussians* rely on pose and point cloud initialization from sharp images, which is an unrealistic assumption. In real-world scenarios, obtaining pose and point cloud initializations directly from extremely blurred images is not possible for NeRF and 3DGS based deblurring methods that rely on COLMAP for initialization. Therefore, we exclude direct comparisons with ExBluRF* and BAD-Gaussians* on real dataset. However, we consider event-based methods as EDI deblurred images from events enable COLMAP initialization even under severe motion blur. We evaluate our results using four standard metrics: PSNR and SSIM for measuring image reconstruction quality, LPIPS for perceptual similarity to the ground truth, and Absolute Pose Error (APE) for assessing the accuracy of estimated camera poses. Higher PSNR and SSIM values indicate better image quality, while lower LPIPS and APE values indicate better perceptual similarity and pose accuracy, respectively.

**Implementation Details:** Our method is implemented in PyTorch Paszke et al. (2019) within the 3DGS-MCMC Kheradmand et al. (2025) framework using the *gsplat* Ye et al. (2025) pipeline. We optimize both Gaussians and camera poses in $SE(3)$ Bézier space using the Adam optimizer. For Bézier, we use 9 control points. The learning rate for Gaussians follows the original 3DGS Kerbl et al. (2023a), while for camera poses, it is set to $1 \times 10^{-3}$. We set the number of virtual camera poses ($n$ in Eq. 2) to 15, ensuring a balance between performance and efficiency. We use 13 event bins for event-based deblurring (EDI). All experiments are conducted on an NVIDIA RTX 4090 GPU with a data factor of 2, using 7k iterations for all experiments and comparisons.

## 3.2 Quantitative Results

**Reconstruction Quality:** Table 1 organizes methods into three categories for fair comparison: (1) those using only motion-blurred images (*w/o Events*), (2) those leveraging event data as additional input (*w/ Events*), and (3) methods requiring sharp images for SfM initialization (*w/ Sharp Supervision*), which are included only for reference due to their impracticality in real-world severe blur. Within this framework, our method achieves superior reconstruction quality across all relevant metrics, PSNR, SSIM, and LPIPS.

**Table 1: Quantitative comparisons for sharp novel view synthesis (deblurring + view synthesis) on the Synthetic Dataset.** The table is organized into three groups for fair comparison: (1) Methods using only motion-blurred images (*w/o Events*), (2) Methods using event data as additional input (*w/ Events*), and (3) Methods requiring sharp images for SfM initialization (*w/ Sharp Supervision*). To ensure fairness, metric rankings are reported separately for Groups 1 and 2. Group 3 methods, such as ExBluRF* and BAD-Gaussians*, are included only for reference and are not ranked, as they rely on sharp images and hence are not practical. Best and second-best results within each ranked group are highlighted in green and orange, respectively.

| Scene | Metric | w/o Events | | | w/ Events | | | | w/ Sharp Supervision(*) | |
|---|---|---|---|---|---|---|---|---|---|---|
| | | MPRNet | Restormer | GeMS (Ours) | EDI+3DGS | E2NeRF | EBAD-NeRF | GeMS-E (Ours) | ExBluRF* | BAD-Gaussians* |
| **Bench** | PSNR↑ | 25.35 | 26.39 | 29.86 | 28.95 | 25.41 | 28.15 | 33.55 | 31.93 | 32.54 |
| | SSIM↑ | 0.678 | 0.720 | 0.841 | 0.865 | 0.708 | 0.822 | 0.924 | 0.877 | 0.901 |
| | LPIPS↓ | 0.425 | 0.356 | 0.118 | 0.201 | 0.438 | 0.172 | 0.063 | 0.111 | 0.046 |
| **Camellia** | PSNR↑ | 24.84 | 25.14 | 28.56 | 22.46 | 28.07 | 24.33 | 29.47 | 28.02 | 28.83 |
| | SSIM↑ | 0.669 | 0.690 | 0.821 | 0.762 | 0.721 | 0.743 | 0.873 | 0.715 | 0.815 |
| | LPIPS↓ | 0.395 | 0.351 | 0.129 | 0.271 | 0.329 | 0.192 | 0.108 | 0.313 | 0.099 |
| **Dragon** | PSNR↑ | 29.96 | 28.37 | 32.43 | 33.27 | 30.89 | 33.99 | 37.01 | 33.45 | 36.98 |
| | SSIM↑ | 0.731 | 0.704 | 0.818 | 0.842 | 0.697 | 0.864 | 0.925 | 0.828 | 0.930 |
| | LPIPS↓ | 0.454 | 0.465 | 0.171 | 0.243 | 0.433 | 0.202 | 0.069 | 0.180 | 0.045 |
| **Jars** | PSNR↑ | 25.36 | 25.57 | 31.42 | 28.13 | 29.85 | 28.89 | 32.35 | 30.85 | 31.52 |
| | SSIM↑ | 0.680 | 0.687 | 0.879 | 0.831 | 0.775 | 0.838 | 0.898 | 0.840 | 0.867 |
| | LPIPS↓ | 0.406 | 0.371 | 0.127 | 0.238 | 0.334 | 0.198 | 0.108 | 0.156 | 0.078 |
| **Jars2** | PSNR↑ | 24.33 | 26.43 | 28.14 | 24.74 | 27.71 | 27.39 | 28.79 | 30.89 | 28.94 |
| | SSIM↑ | 0.745 | 0.814 | 0.873 | 0.812 | 0.770 | 0.863 | 0.906 | 0.860 | 0.851 |
| | LPIPS↓ | 0.358 | 0.275 | 0.173 | 0.262 | 0.383 | 0.171 | 0.133 | 0.113 | 0.114 |
| **Postbox** | PSNR↑ | 25.89 | 26.52 | 27.74 | 24.99 | 30.66 | 26.82 | 31.33 | 31.40 | 26.40 |
| | SSIM↑ | 0.736 | 0.753 | 0.788 | 0.789 | 0.813 | 0.826 | 0.906 | 0.864 | 0.757 |
| | LPIPS↓ | 0.318 | 0.286 | 0.150 | 0.228 | 0.262 | 0.151 | 0.070 | 0.095 | 0.123 |
| **Stone Lantern** | PSNR↑ | 24.97 | 26.68 | 28.29 | 26.48 | 30.47 | 26.29 | 29.43 | 28.24 | 28.29 |
| | SSIM↑ | 0.785 | 0.831 | 0.849 | 0.825 | 0.836 | 0.802 | 0.894 | 0.765 | 0.843 |
| | LPIPS↓ | 0.342 | 0.280 | 0.195 | 0.270 | 0.324 | 0.264 | 0.152 | 0.236 | 0.143 |
| **Sunflowers** | PSNR↑ | 28.86 | 29.55 | 29.47 | 31.38 | 31.74 | 30.98 | 33.69 | 34.46 | 34.06 |
| | SSIM↑ | 0.837 | 0.847 | 0.854 | 0.914 | 0.850 | 0.903 | 0.938 | 0.920 | 0.942 |
| | LPIPS↓ | 0.242 | 0.206 | 0.163 | 0.144 | 0.310 | 0.117 | 0.077 | 0.093 | 0.065 |
| **Average** | PSNR↑ | 26.19 | 26.83 | 29.49 | 27.55 | 29.35 | 28.36 | 31.95 | 31.15 | 30.95 |
| | SSIM↑ | 0.733 | 0.756 | 0.840 | 0.830 | 0.771 | 0.833 | 0.908 | 0.834 | 0.863 |
| | LPIPS↓ | 0.368 | 0.324 | 0.153 | 0.232 | 0.352 | 0.183 | 0.097 | 0.162 | 0.089 |

GeMS, which operates solely on motion-blurred images without event data, outperforms all competing methods in its group and even surpasses event-based approaches, demonstrating robustness to extreme blur. When event data is available, GeMS-E further improves performance, achieving an average 2.5 dB PSNR gain over the state-of-the-art event-based method E2NeRF and most significantly, our GeMS-E delivers a 1 dB PSNR improvement over sharp-supervised baselines (ExBluRF*, BAD-Gaussians*) despite their privileged access to sharp images. This performance gap is particularly notable given that sharp-supervised

methods rely on impractical initialization. The integration of event-based deblurring, VGGSfM-based SfM initialization, and 3DGS-MCMC joint optimization enables GeMS-E to deliver high-fidelity reconstructions with significantly lower computational cost. Overall, our approach provides a practical and effective solution for reliable 3D reconstruction in severely motion-blurred scenarios. Results on deblurring (Table 7) are presented in Appendix.

**GeMS-E Outperforms Sharp-Supervised Baselines BAD-Gaussians\* & ExBluRF\*:** GeMS-E achieves superior performance primarily due to its highly accurate initialization and advanced motion modeling. Event-based deblurring (EDI) produces sharp-enough images for VGGSfM to estimate camera poses that are very close to ground truth (mean APE: 0.0862), which is crucial for reliable 3D reconstruction under extreme motion blur. Compared to BAD-Gaussians, GeMS-E provides robust probabilistic refinement through 3DGS-MCMC, which adaptively densifies geometry using noise-aware sampling and is more resilient to pose errors than the heuristic-based approach in BAD-Gaussians. Additionally, GeMS-E's joint optimization of Bézier trajectories and Gaussian parameters ensures physical consistency, whereas BAD-Gaussians' linear or spline motion approximation can introduce geometric inaccuracies, as presented in Figure 12. In contrast to ExBluRF, GeMS-E excels at modeling complex, continuous camera motion: while ExBluRF's voxel-based representation struggle with extreme motion blur, GeMS-E leverages a bundle-adjusted radiance field representation enabling higher-quality reconstructions. This combination of accurate initialization using VGGSfM from EDI motion-blurred images and 3DGS-MCMC joint optimization with sophisticated motion modeling enables GeMS-E to consistently outperform both ExBluRF and BAD-Gaussians, even when those methods have access to sharp image supervision.

**Training Time and GPU Memory Consumption:** We evaluate the efficiency of our approach against state-of-the-art event-based methods, EBAD-NeRF and E2NeRF, in terms of training time and GPU memory usage on real dataset. As shown in Table 5, our method significantly reduces training time, completing optimization in approximately **7 minutes** per scene, whereas EBAD-NeRF and E2NeRF require over 6 hours and 14 hours, respectively. Moreover, Table 6 highlights the GPU memory consumption across different scenes, where our approach requires only $\sim$**1.55 GiB** on an average, compared to the excessive demands of EBAD-NeRF ($\sim$14.49 GiB) and E2NeRF ($\sim$15.16 GiB). These results confirm the scalability and hardware efficiency of our method, making it practical for real-world applications.

### 3.3 Qualitative Results

We present qualitative comparisons for deblurring on both synthetic and real datasets in Figure 4 and Figure 5, respectively. Novel view synthesis results (Figure 14) are presented in Appendix. Our method GeMS-E consistently outperforms existing approaches, producing sharper reconstructions with fewer artifacts and improved texture details. In synthetic scenes such as *Stone Lantern* and *Jars*, competing methods struggle to recover fine structures, often leading to over-smoothed or distorted reconstructions, as indicated by the red arrows in Figure 4. In contrast, our approach faithfully restores object details and edges. Similarly, in real-world datasets as shown in Figure 5, our method achieves superior clarity, particularly in challenging regions with high-frequency textures, such as text on the *CVPR 2023* poster or specular highlights on metallic surfaces. Unlike prior event-based methods, such as E2NeRF and EBAD-NeRF, which introduce color distortions or fail to fully remove motion blur, our approach effectively preserves accurate color distributions while restoring sharp details. Moreover, our method demonstrates improved robustness in handling fine-grained textures, such as plant leaves and intricate object boundaries, where others exhibit blurring or ghosting artifacts. This advantage is evident in diverse scenes, reinforcing the effectiveness of our framework. Additional qualitative results of our methods, GeMS and GeMS-E, are presented in appendix E. The supplementary material provides novel sharp free-viewpoint renderings for both synthetic and real-world scenes.

### 3.4 Ablations

To comprehensively evaluate our framework, we first assess the robustness of each module across progressive motion blur levels, quantifying their effectiveness under varying degrees of blur. We then conduct targeted

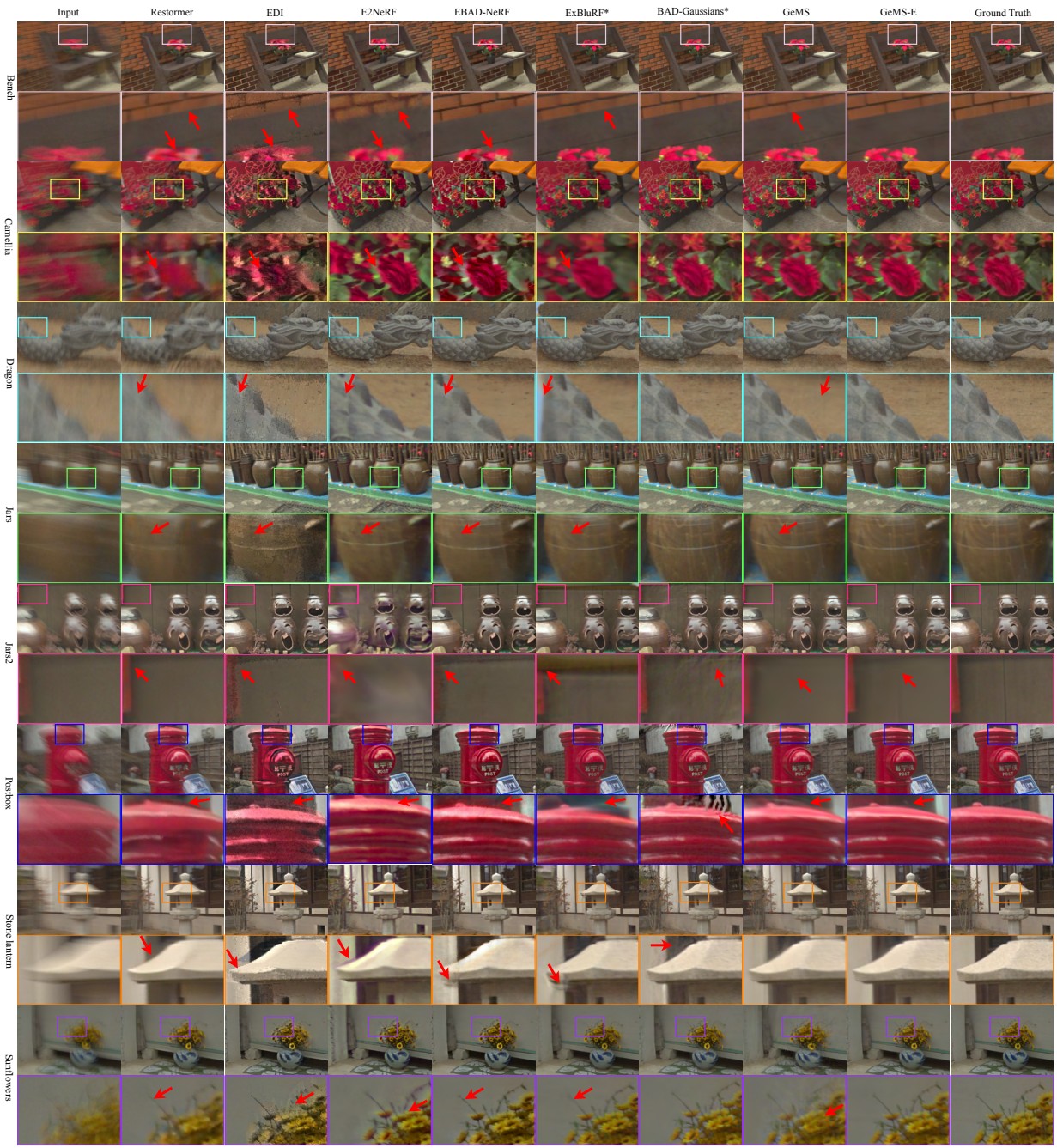

**Figure 4: Results on the Synthetic Dataset:** Our method effectively removes severe motion blur, reconstructing sharp results with high fidelity. Compared to existing approaches, it better preserves fine details and structural consistency while reducing color artifacts, demonstrating robustness under extreme blur conditions. Note that ExBluRF* and BAD-Gaussians* rely on pose and point cloud initializations from sharp images.

analyses in extreme blur scenarios, specifically validating the performance and individual contributions of each module.

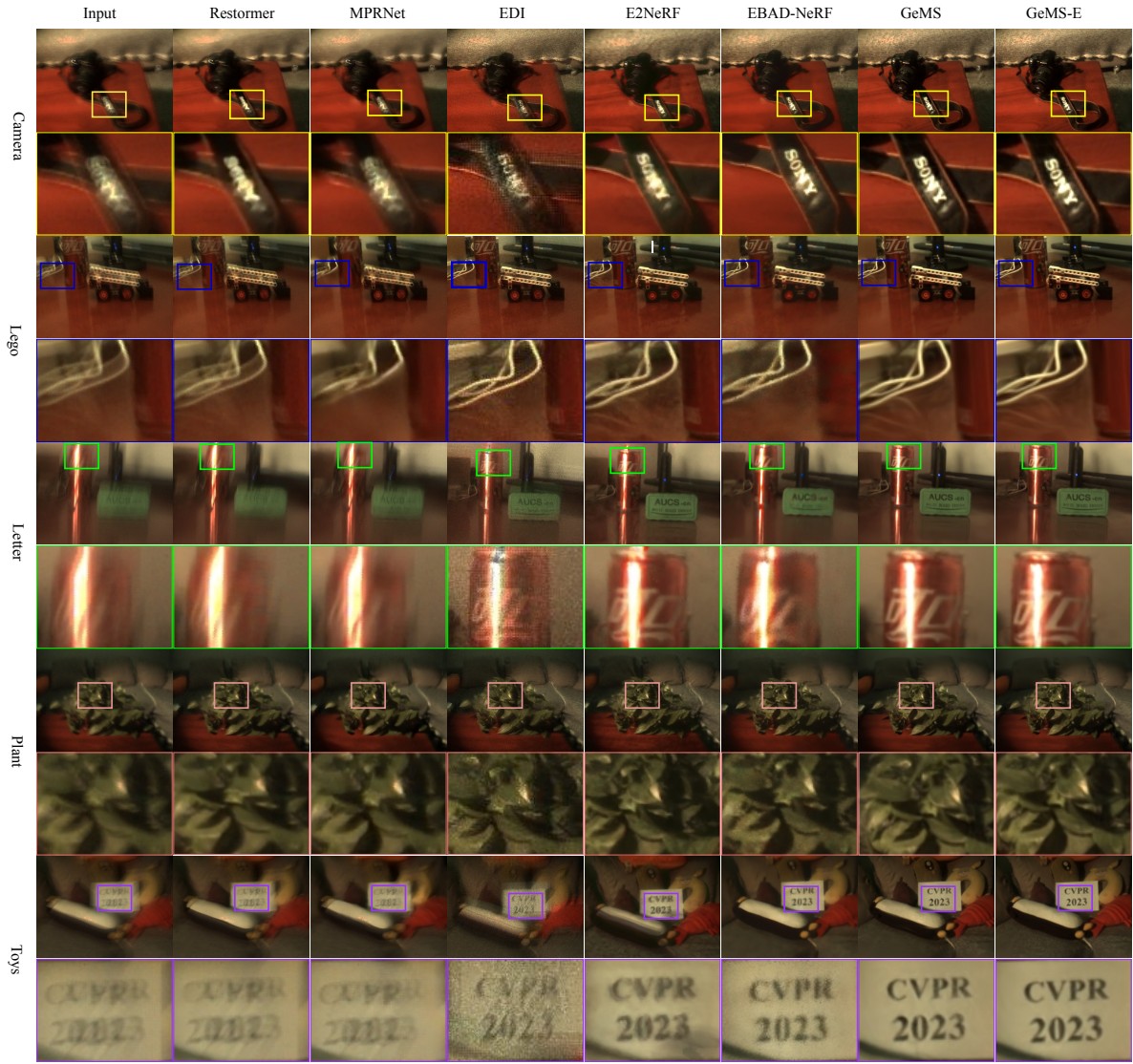

**Figure 5: Results on the Real Dataset:** Our method reconstructs sharp and high-quality images from severely motion-blurred real-world inputs. In contrast, existing methods struggle with artifacts, noise, loss of fine details, and text degradation. Our framework effectively restores textures and structural consistency, as evident in the insets.

### 3.4.1 Robustness of Modules across various Blur Levels

**VGGSfM Robustness:** To systematically evaluate VGGSfM's robustness to motion blur, we designed a controlled experiment using multi-frame averaging to simulate increasing blur severity. Starting with a burst sequence of 11 sharp images per viewpoint, we created progressively blurred inputs by averaging 1, 3, 5, 7, 9, and 11 consecutive frames (Figure 6). Each blurred image was processed through both VGGSfM and COLMAP. While COLMAP failed to register images beyond moderate blur levels, VGGSfM consistently registered all images across all blur levels, producing stable camera poses and dense point clouds even under extreme blur (Table 2). To further evaluate VGGSfM's effectiveness, we plotted the translational (x, y, z) and rotational (yaw, pitch, roll) poses of the blurry reconstructions with respect to the sharp ones, as well as the translational pose errors relative to ground truth for each blur level (Figure 7 & Table 3). This clearly demonstrates that VGGSfM remains robust and reliable even in extreme motion blur scenarios where traditional methods fail.

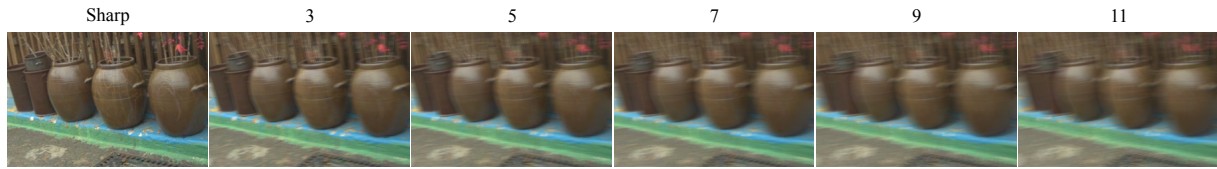

| Sharp | 3 | 5 | 7 | 9 | 11 |

**Figure 6: Synthetic Motion Blur Levels:** Images generated by averaging 1 (sharp), 3, 5, 7, 9, and 11 consecutive frames from an 11-image burst at each viewpoint, illustrating the increasing severity of motion blur used for evaluation.

**Table 2:** SfM comparison: Performance of VGGSfM vs COLMAP across various blur levels.

| Blur | VGGSfM | | COLMAP | |
|------|--------|--------|--------|--------|
| | #Pts | #Imgs | #Pts | #Imgs |
| Sharp | 23498 | 20 | 4987 | 20 |
| 3 | 22873 | 20 | 1257 | 20 |
| 5 | 21660 | 20 | 606 | 20 |
| 7 | 18518 | 20 | 246 | 15 |
| 9 | 21351 | 20 | x | x |
| 11 | 20283 | 20 | x | x |

**Table 3:** VGGSfM Pose translation error statistics (in meters) across blur levels with respect to ground truth.

| Blur | RMSE ↓ | Mean ↓ | Med. ↓ | Std ↓ | Max ↓ |
|------|--------|--------|--------|-------|-------|
| 3 | 0.18 | 0.16 | 0.13 | 0.09 | 0.35 |
| 5 | 0.32 | 0.27 | 0.20 | 0.18 | 0.65 |
| 7 | 0.40 | 0.36 | 0.29 | 0.17 | 0.75 |
| 9 | 0.61 | 0.49 | 0.36 | 0.36 | 1.43 |
| 11 | 0.72 | 0.52 | 0.35 | 0.50 | 2.24 |

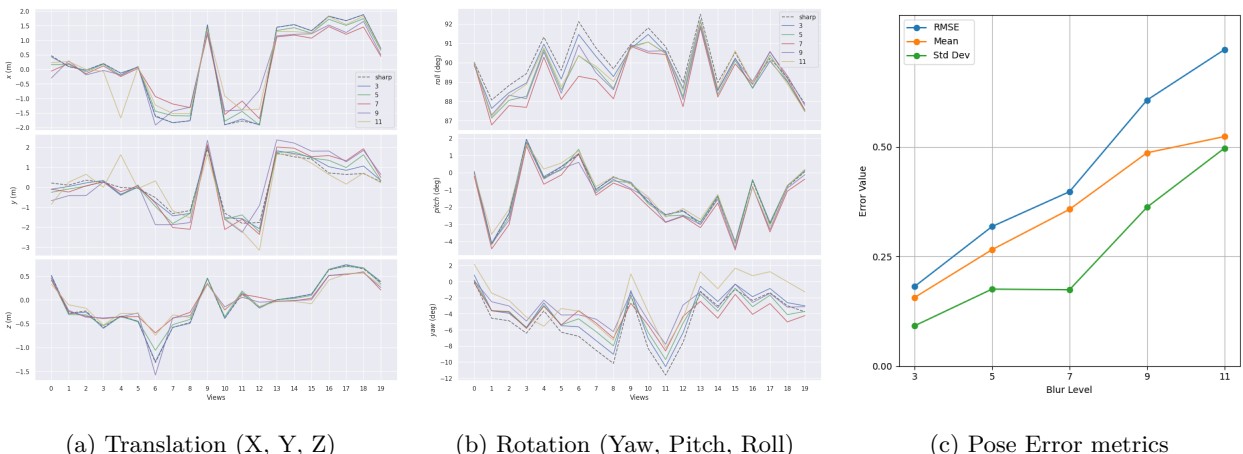

(a) Translation (X, Y, Z)    (b) Rotation (Yaw, Pitch, Roll)    (c) Pose Error metrics

**Figure 7: Robustness of VGGSfM Pose Estimation across Blur Levels:** Evaluation of VGGSfM's pose estimation performance using sharp versus motion-blurred inputs. (a) Estimated translation parameters (X,Y,Z) for 20 views compared to ground truth. (b) Estimated rotation parameters (yaw, pitch, roll) for corresponding views versus ground truth. (c) Absolute pose error statistics for each blur level relative to ground truth.

**3DGS-MCMC Robustness:** To systematically evaluate the robustness of 3DGS-MCMC to blur-corrupted initializations, we generated point clouds from images with varying levels of motion blur using VGGSfM, resulting in increasingly noisy and sparse initial geometry (e.g., blur-3, blur-5).

We conducted two systematic experiments to thoroughly evaluate the robustness of 3DGS-MCMC under varying levels of motion blur. In the first experiment, we focused on the effect of blur-corrupted initializations by generating point clouds from images with different degrees of motion blur using VGGSfM, resulting in increasingly noisy and sparse geometry. These blur-degraded point clouds were then used as the starting input for both standard 3DGS and 3DGS-MCMC, with the reconstruction loss computed against sharp ground truth images. The results, as shown in Figure 8 and Table 8, clearly demonstrate that standard 3DGS suffers significant performance degradation as blur increases, whereas 3DGS-MCMC maintains stable reconstruction quality across all blur levels. This quantitatively validates that the probabilistic sampling framework of MCMC is inherently robust to initialization quality, and specifically effective for blur-corrupted point cloud initializations.

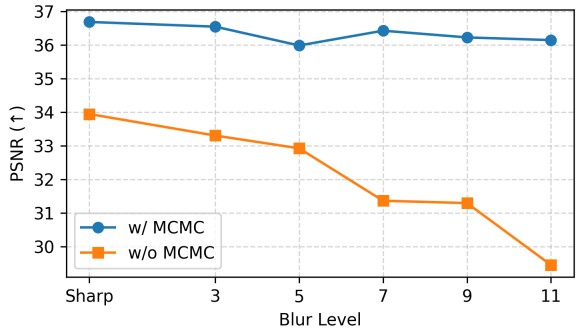 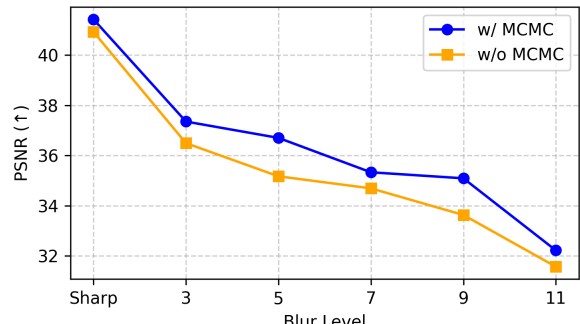

**Figure 8: MCMC robustness to various blur-corrupted point cloud initializations:** PSNR comparison of 3DGS-MCMC with 3DGS across various blur point cloud initializations obtained from VGGSfM.

**Figure 9: MCMC robustness to various blur levels in GeMS:** PSNR comparison of GeMS for deblurring with and without MCMC across various blur levels, SfM initialization are obtained from VGGSfM.

In the second experiment, we assessed the deblurring performance within our GeMS framework by comparing GeMS with MCMC-based probabilistic refinement against a variant of GeMS without MCMC. Both versions were initialized directly on motion-blurred images and optimized end-to-end using photometric loss against the original blurry images, across a range of blur levels. As reported in Figure 9 and Table 9, GeMS with MCMC consistently maintains better reconstruction quality across increasing blur severity. Furthermore, our ablation results (Figure 10, Figure 11) reveal that the non-MCMC variant exhibits noticeable artifacts and structural inconsistencies in extreme motion blur scenarios, highlighting the practical value of MCMC for reliable scene recovery under challenging extreme motion blur conditions.

### 3.4.2 Component-wise Analysis for Extreme Motion Blur

To quantify the contribution of each module (VGGSfM, MCMC, EDI) in GeMS and GeMS-E for extreme motion deblurring, we performed ablation studies on the synthetic dataset, with results summarized in Table 4.

**Table 4: Ablation study for novel sharp view synthesis (deblurring + novel view synthesis) on the synthetic dataset.** We evaluate the impact of different components, VGGSfM, MCMC and EDI on novel view synthesis performance. The inclusion of each module leads to considerable improvements in PSNR, SSIM, and LPIPS, with their combination yielding the best overall results, demonstrating the synergistic benefits of our approach. Abbreviations used for the ablation methods are outlined in appendix D.3.

| Method | Metric | Bench | Camellia | Dragon | Jars | Jars2 | Postbox | Stone L. | Sunflowers | Average |
|---|---|---|---|---|---|---|---|---|---|---|
| w/o MCMC + w/o EDI + w/ VGGSfM | PSNR↑ | 30.06 | 27.80 | 33.19 | 30.89 | 28.51 | 25.86 | 27.22 | 28.64 | 29.02 |
| | SSIM↑ | 0.832 | 0.772 | 0.831 | 0.842 | 0.838 | 0.688 | 0.821 | 0.827 | 0.806 |
| | LPIPS↓ | 0.097 | 0.118 | 0.081 | 0.097 | 0.102 | 0.177 | 0.169 | 0.226 | 0.133 |
| w/ MCMC + w/o EDI + w/ HLOC | PSNR↑ | 30.32 | 27.07 | 31.01 | 28.62 | 27.85 | 22.94 | 24.73 | 31.76 | 28.04 |
| | SSIM↑ | 0.838 | 0.717 | 0.773 | 0.752 | 0.816 | 0.586 | 0.706 | 0.898 | 0.761 |
| | LPIPS↓ | 0.101 | 0.154 | 0.118 | 0.146 | 0.104 | 0.218 | 0.251 | 0.090 | 0.148 |
| **GeMS (w/ MCMC + w/o EDI + w/ VGGSfM)** | PSNR↑ | 29.86 | 28.56 | 32.43 | 31.42 | 28.14 | 27.74 | 28.29 | 29.47 | 29.49 |
| | SSIM↑ | 0.841 | 0.821 | 0.818 | 0.879 | 0.873 | 0.788 | 0.849 | 0.854 | 0.840 |
| | LPIPS↓ | 0.118 | 0.129 | 0.171 | 0.127 | 0.173 | 0.150 | 0.195 | 0.163 | 0.153 |
| w/o MCMC + w/ EDI + w/ VGGSfM | PSNR↑ | 32.63 | 28.70 | 36.41 | 31.66 | 28.60 | 28.52 | 27.97 | 33.97 | 31.06 |
| | SSIM↑ | 0.901 | 0.811 | 0.933 | 0.865 | 0.839 | 0.803 | 0.855 | 0.942 | 0.869 |
| | LPIPS↓ | 0.042 | 0.101 | 0.039 | 0.084 | 0.104 | 0.085 | 0.145 | 0.069 | 0.083 |
| w/ MCMC + w/ EDI + w/ COLMAP | PSNR↑ | 32.33 | 30.05 | 35.03 | 31.44 | 28.93 | 30.67 | 28.29 | 33.51 | 31.28 |
| | SSIM↑ | 0.914 | 0.872 | 0.866 | 0.876 | 0.886 | 0.887 | 0.852 | 0.933 | 0.886 |
| | LPIPS↓ | 0.081 | 0.111 | 0.203 | 0.147 | 0.161 | 0.098 | 0.224 | 0.086 | 0.139 |
| **GeMS-E (w/ MCMC + w/ EDI + w/ VGGSfM)** | PSNR↑ | 33.55 | 29.47 | 37.01 | 32.35 | 28.79 | 31.33 | 29.43 | 33.69 | 31.95 |
| | SSIM↑ | 0.924 | 0.873 | 0.925 | 0.898 | 0.906 | 0.906 | 0.894 | 0.938 | 0.908 |
| | LPIPS↓ | 0.063 | 0.108 | 0.069 | 0.108 | 0.133 | 0.070 | 0.152 | 0.077 | 0.097 |

Our analysis shows that COLMAP fails entirely under severe motion blur, while HLOC, although able to run, lags behind VGGSfM by an average of 1.45 dB in PSNR, highlighting VGGSfM's superior robustness in challenging conditions. The inclusion of MCMC-based probabilistic refinement further improves reconstruction quality, contributing an additional 0.47 dB in PSNR and noticeably reducing artifacts, as seen in Figure 10. Event-based deblurring (EDI) provides the most significant boost, with its integration yielding a 2.46 dB PSNR improvement over GeMS and producing the sharpest and most faithful reconstructions, particularly in extreme blur scenarios. The full GeMS-E pipeline, combining VGGSfM, MCMC, and EDI, achieves the highest overall performance, as evidenced by both quantitative results and qualitative examples in Figure 11. For further details on ablation method abbreviations and corresponding results, please refer to the appendix (appendix D.3).

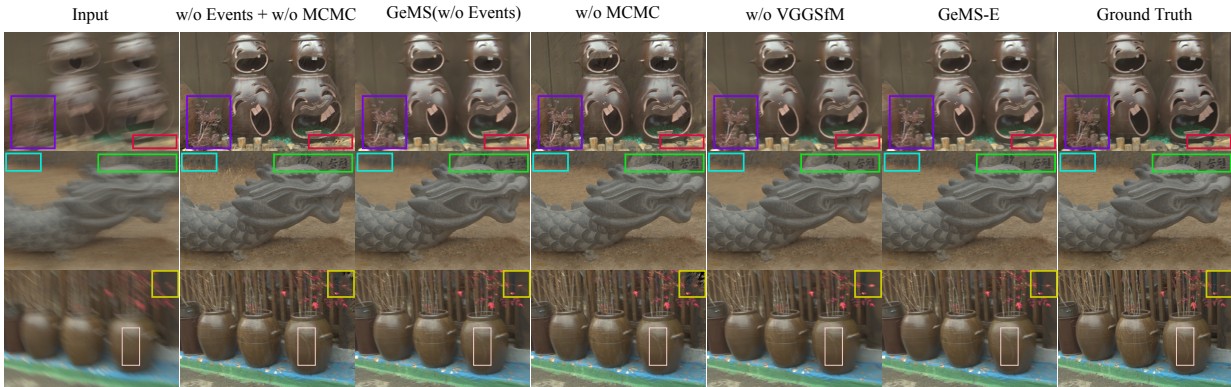

**Figure 10: Ablations on different modules of our framework on the Synthetic Dataset:** Our method (GeMS) achieves strong deblurring with MCMC helping to reduce artifacts and enhance reconstruction quality. With events (GeMS-E), the results become even sharper and more detailed, demonstrating the added benefit of event information.

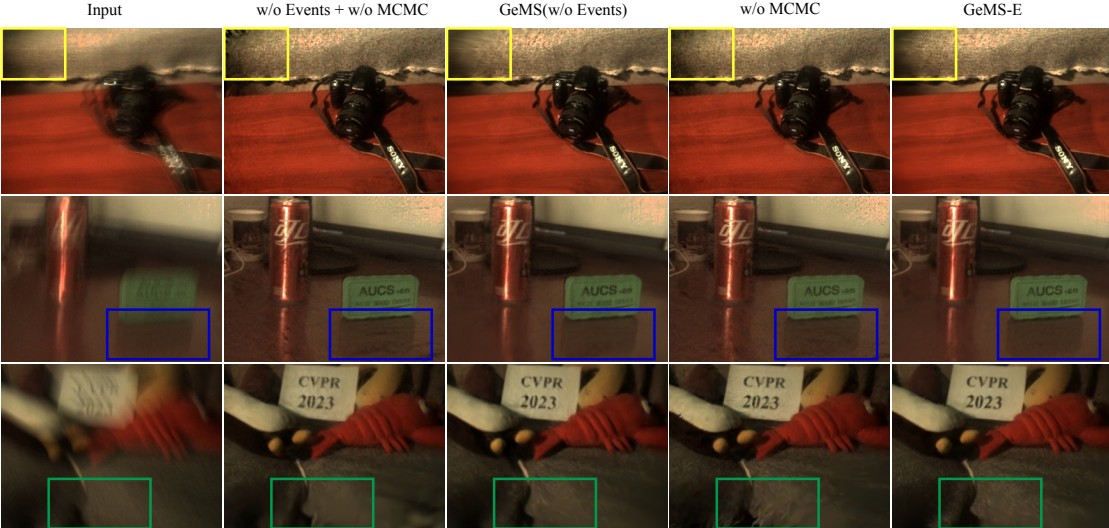

**Figure 11: Ablation study on different modules of our framework on the Real Dataset:** GeMS produces high-quality reconstructions, with MCMC effectively suppressing artifacts. Incorporating event data in GeMS-E further refines the results, demonstrating the effectiveness of our approach across different blur settings.

**Table 5: Training Time Comparison on the Real Dataset (hh:mm:ss):** Comparison of training times for different methods on real datasets, highlighting the efficiency of our approach.

|  | Camera↓ | Lego↓ | Letter↓ | Plant↓ | Toys↓ | Avg↓ |
|---|---|---|---|---|---|---|
| EBAD-NeRF | 06:44:00 | 06:44:00 | 06:44:00 | 06:44:00 | 06:44:00 | 06:44:00 |
| E2NeRF | 14:58:00 | 14:58:00 | 14:58:00 | 14:58:00 | 14:58:00 | 14:58:00 |
| **Ours** | 00:07:14 | 00:07:23 | 00:07:18 | 00:06:58 | 00:07:11 | 00:07:13 |

**Table 6: GPU Memory Usage on the Real Dataset (MiB):** Evaluation of GPU memory consumption across different methods, demonstrating the significant reduction in GPU memory usage.

|  | Camera↓ | Lego↓ | Letter↓ | Plant↓ | Toys↓ | Avg↓ |
|---|---|---|---|---|---|---|
| EBAD-NeRF | 14836 | 14836 | 14836 | 14836 | 14836 | 14836 |
| E2NeRF | 15532 | 15532 | 15532 | 15532 | 15532 | 15532 |
| **Ours** | 1571 | 1897 | 1730 | 1341 | 1382 | 1584 |

### 3.4.3 Number of Virtual Cameras & Trajectory Representations

We conducted experiments to analyze the impact of the number of interpolated virtual camera poses within the duration of exposure, denoted as $n$, and the effectiveness of different motion trajectory representations. We varied $n$ across $\{5, 10, 15, 20\}$, with the corresponding rendering results summarized in Figure 12. Our findings indicate that increasing $n$ up to a certain threshold effectively mitigates severe motion blur, beyond which performance begins to decline; hence, we adopt $n = 15$. Additionally, we conducted ablations using linear interpolation, cubic B-splines, and Bézier curves for trajectory representations, with results also summarized in Figure 12. Our findings demonstrate that Bézier curves outperform other methods, as they better capture the complex, non-uniform motion present in heavily blurred scenes. Bézier curves consistently achieve superior performance across all metrics in severe motion blur scenarios, providing smoother and more accurate trajectory representations, making them the optimal choice for our framework.

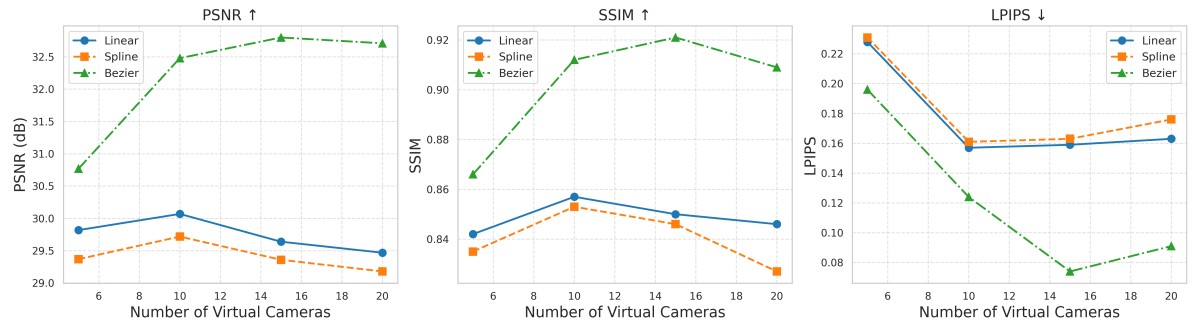

**Figure 12: Impact of trajectory representations and the number of virtual cameras:** Comparison of Linear, Spline, and Bezier trajectory representations across different virtual camera counts. Bezier interpolation consistently performs better than Linear and Spline representations, demonstrating its advantage in generating high-quality sharp novel views.

## 4 Conclusion

In this work, we introduced GeMS, an efficient 3D Gaussian Splatting framework that reconstructs sharp 3D scenes directly from severely motion-blurred images. By integrating VGGSfM for blur-robust initialization, 3DGS-MCMC for probabilistic scene modeling, and joint trajectory-geometry optimization, our approach forms a tightly coupled, end-to-end differentiable pipeline where each component compensates for the limitations of the others. This synergy enables mutual refinement: VGGSfM's initialization is adaptively densified by 3DGS-MCMC, while joint optimization continuously aligns scene geometry and camera motion using a physics-based blur image formation model. When event data is available, we extend this framework with GeMS-E by using the Event-based Double Integral (EDI) model to deblur the inputs; these deblurred images are then passed through our GeMS framework for further refinement, especially in scenarios where all input images are severely motion blurred. Extensive experiments on both synthetic and real-world datasets demonstrate that GeMS and GeMS-E consistently outperform state-of-the-art methods in accuracy, efficiency, and robustness, even under extreme motion blur. Our work establishes a new paradigm for motion-robust 3D reconstruction, moving beyond the limitations of COLMAP and enabling reliable scene recovery in extreme motion blur scenarios previously considered intractable. Future work will explore de-

blurring with an extremely sparse set of images, ultimately pushing the framework's limits to handle the most challenging scenario: reconstructing sharp scene from a single motion-blurred image under extreme motion blur. Additionally, we aim to extend GeMS / GeMS-E to dynamic scenes.

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

## Appendix

In the appendix, we provide additional information as outlined below:

- appendix A presents an overview of related works.

- appendix B details the background of 3D Gaussian Splatting.

- appendix C elaborates on the EDI model.

- appendix D presents additional quantitative results and ablations from experimental section.

- appendix E presents additional qualitative results of our methods, GeMS and GeMS-E, on both synthetic and real datasets across two different views.

## A   Related Work

### A.1   SfM for NeRF/3DGS Initialization

Accurate camera pose estimation and sparse 3D reconstruction are essential for initializing NeRF and 3D Gaussian Splatting (3DGS). COLMAP Schonberger & Frahm (2016) is the most widely used incremental SfM pipeline, but its reliance on SIFT-based feature matching makes it vulnerable to motion blur and low-texture failures. GLOMAP Pan et al. (2024) improves scalability with a global SfM approach, while HLOC Sarlin et al. (2019) enhances wide-baseline localization using SuperGlue Sarlin et al. (2020). However, both methods remain dependent on pairwise feature correspondences, limiting their effectiveness under extreme motion blur. To address this, Pixel-Perfect SfM Lindenberger et al. (2021) refines both keypoint locations and camera poses by optimizing a featuremetric error using dense deep features. This improves the geometric accuracy of SfM pipelines, making it more robust to detection noise and appearance variations. Further advancing SfM, VGGSfM Wang et al. (2024) replaces traditional keypoint matching with deep 2D point tracking, eliminating the need for pairwise matching. It jointly optimizes camera poses and 3D points via a fully differentiable bundle adjustment layer, achieving state-of-the-art performance on CO3D Reizenstein et al. (2021), IMC Phototourism Jin et al. (2021), and ETH3D Schops et al. (2017). Given its superior robustness in extreme motion blur scenarios, VGGSfM is the preferred SfM method for initializing 3DGS in our deblurring pipeline.

### A.2   Novel View Synthesis

NeRF Mildenhall et al. (2020) has garnered significant attention in 3D vision due to its remarkable ability to generate photo-realistic novel views. At its core, NeRF employs a neural implicit representation optimized via differentiable volume rendering. Numerous works have sought to enhance its rendering quality Barron et al. (2021; 2022); Jiang et al. (2023); Wu et al. (2022); Zhang et al. (2020), while others have focused on accelerating both training and rendering Lindell et al. (2021); Reiser et al. (2021); Yu et al. (2021); Fridovich-Keil et al. (2022); Müller et al. (2022); Chen et al. (2022); Sun et al. (2022), leading to significant improvements in efficiency.

Recently, *3D Gaussian Splatting* (3DGS) Kerbl et al. (2023a) has emerged as an efficient alternative to radiance field models, excelling in both fine-grained scene reconstruction and real-time rendering. By replacing NeRF's computationally expensive ray marching Max (1995) with a deterministic rasterization technique, 3DGS preserves visual fidelity while enabling rapid rendering. However, 3DGS requires *accurate camera poses and point cloud initialization* for effective reconstruction, which can be challenging in motion-blurred scenarios. To further enhance stability and accuracy, recent works have explored *MCMC-based Gaussian Splatting* Kheradmand et al. (2025), treating Gaussians as probabilistic samples for more robust initialization and optimization. Inspired by this, our method integrates *MCMC-based Gaussian Splatting*, improving reconstruction quality in challenging motion-blurred scenarios.

### A.3 Radiance Field Deblurring

NeRF-based approaches have explored various techniques to reconstruct sharp scene representations from motion-blurred images. Deblur-NeRF Ma et al. (2022) and ExBluRF Lee et al. (2023) model the blur process during scene optimization but assume fixed camera poses, which can be inaccurate under severe motion blur. To address this, BAD-NeRF Wang et al. (2023) jointly optimizes camera motion and radiance fields, allowing pose refinement. However, its reliance on implicit MLP-based representations leads to slow optimization and rendering times. Explicit representations, such as 3D Gaussian Splatting, have recently emerged as efficient alternatives for real-time rendering. BAD-Gaussians Zhao et al. (2024) extends 3DGS by jointly optimizing Gaussians and camera trajectories, improving both efficiency and reconstruction quality. Similarly, Deblur-GS Chen & Liu (2024) refines Gaussian parameters to achieve sharper reconstructions but still relies on COLMAP for initialization, making it ineffective in extreme motion blur scenarios.

Event-based methods have also been introduced to tackle motion blur in 3D reconstruction. E2NeRF Qi et al. (2023) and EBAD-NeRF Qi et al. (2024) integrate event streams into NeRF-based frameworks, leveraging high temporal resolution for deblurring. However, they inherit the inefficiencies of NeRF's implicit representations. More recently, E2GS Deguchi et al. (2024) has attempted to incorporate event data into Gaussian Splatting, but it fails to effectively remove motion blur, introducing severe color artifacts. To overcome these challenges, we propose a novel Gaussian Splatting framework that directly processes motion-blurred images while incorporating event-based deblurring in extreme cases where event data is available. By leveraging VGGSfM for robust pose estimation and MCMC-based Gaussian Splatting for adaptive initialization, our approach jointly optimizes camera poses and Gaussian parameters, overcoming the limitations of previous methods and achieving high-quality, real-time 3D reconstruction under severe motion blur.

## B  Background

### B.1  3D Gaussian Scene Representation

In the 3DGS framework Kerbl et al. (2023a), a scene is modeled as a collection of 3D Gaussian distributions, each described by its mean position $\boldsymbol{\mu} \in \mathbb{R}^3$, a 3D covariance matrix $\boldsymbol{\Sigma} \in \mathbb{R}^{3\times3}$, opacity $\mathbf{o} \in \mathbb{R}$, and color $\mathbf{c} \in \mathbb{R}^3$. The distribution of each Gaussian is represented by the following formula:

$$\mathbf{G}(\mathbf{x}) = e^{-\frac{1}{2}(\mathbf{x}-\boldsymbol{\mu})^\top \boldsymbol{\Sigma}^{-1}(\mathbf{x}-\boldsymbol{\mu})} \tag{9}$$

The covariance matrix $\boldsymbol{\Sigma}$ is parameterized by a scale matrix $\mathbf{S} \in \mathbb{R}^3$ and a rotation matrix $\mathbf{R} \in \mathbb{R}^{3\times3}$. The rotation matrix $\mathbf{R}$ is represented using a quaternion $\mathbf{q} \in \mathbb{R}^4$, while the decomposition

$$\boldsymbol{\Sigma} = \mathbf{R}\mathbf{S}\mathbf{S}^T\mathbf{R}^T \tag{10}$$

ensures that $\boldsymbol{\Sigma}$ remains positive definite.

For rendering, 3D Gaussians are projected into 2D space from the camera pose $\mathbf{T}_c = \{\mathbf{R}_c \in \mathbb{R}^{3\times3}, \mathbf{t}_c \in \mathbb{R}^3\}$ using the following equations:

$$\boldsymbol{\Sigma}' = \mathbf{J}\mathbf{R}_c\boldsymbol{\Sigma}\mathbf{R}_c^T\mathbf{J}^T \tag{11}$$

where $\boldsymbol{\Sigma}' \in \mathbb{R}^{2\times2}$ is the 2D covariance matrix and $\mathbf{J} \in \mathbb{R}^{2\times3}$ is the Jacobian matrix for the projection.

To compute the rendered pixel colors, the 2D Gaussians are rasterized based on their depth values:

$$\mathbf{C} = \sum_{i=1}^{N} \mathbf{c}_i \alpha_i \prod_{j=1}^{i-1}(1-\alpha_j) \tag{12}$$

where $\mathbf{c}_i$ is the color of the $i$-th Gaussian, and $\alpha_i$ is the corresponding alpha value:

$$\alpha_i = \mathbf{o}_i \cdot \exp(-\sigma_i), \quad \sigma_i = \frac{1}{2}\Delta_i^T \boldsymbol{\Sigma}'^{-1}\Delta_i \tag{13}$$

Here, $\Delta_i \in \mathbb{R}^2$ represents the distance between the pixel center and the center of the 2D Gaussian. The pixel color $\mathbf{C}$ is differentiable with respect to both the Gaussian parameters $\mathbf{G}$ and the camera pose $\mathbf{T}_c$.

## C Method

### C.1 EDI (Event Double Integral)

The event-based double integral model (EDI) Pan et al. (2019) reconstructs multiple sharp images from a single motion-blurred image using event data. Given a blurred image $I_{\text{blur}}$ and a sequence of event bins $\{B_k\}_{k=1}^b$, the simplified EDI formulation adopted by E2NeRF Qi et al. (2023) models event accumulation to estimate sharp images over time.

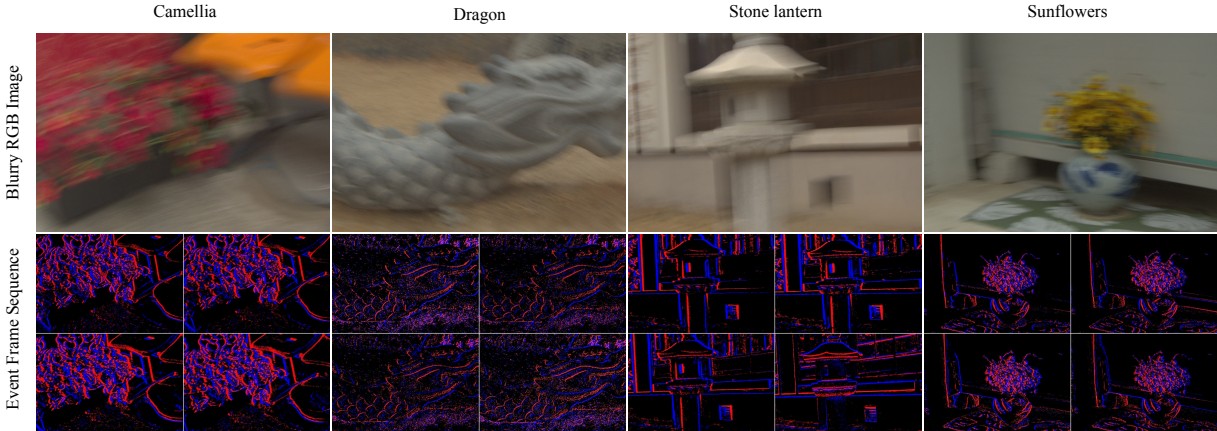

**Figure 13: Our Complementary Event Dataset (EveGeMS):** We introduce a synthetic event-based dataset tailored for cases with intense motion blur. Despite the heavy degradation in the RGB frames, each is paired with a stream of event frames that capture precise structural and motion details. This complementary signal enables reliable deblurring and novel view reconstruction, even when all input views are significantly blurred.

We define the accumulated event count up to bin $k$ as:

$$S_k = \sum_{i=1}^{k} B_i \tag{14}$$

The event-compensated image intensity is iteratively computed as:

$$E_k = E_{k-1} + e^{\Theta S_k}, \quad E_0 = 1, \quad k = 1, 2, \ldots, b \tag{15}$$

where $\Theta$ is the event threshold controlling the influence of accumulated events.

The initial reference sharp image $I_0$ is estimated as:

$$I_0 = \frac{(b+1)I_{\text{blur}}}{E_b} \tag{16}$$

where $E_b$ corresponds to the final accumulated intensity estimate.

Each deblurred image $I_k$ is obtained by propagating the initial image using event accumulation:

$$I_k = I_0 e^{\Theta S_k}, \quad k = 1, 2, \ldots, b \tag{17}$$

Among the reconstructed deblurred images $\{I_k\}_{k=0}^b$, we use the middle frame as a reference for the corresponding blurry image. These reference deblurred images are then fed into our GeMS framework. Using event data for initialization, *GeMS-E* enhances robustness against extreme motion blur, enabling reliable 3D reconstruction even when all input views are severely blurred.

# D   Experiments

## D.1   Implementation Details:

Our method is implemented in PyTorch Paszke et al. (2019) within the 3DGS-MCMC Kheradmand et al. (2025) framework using the *gsplat* Ye et al. (2025) pipeline. We optimize both Gaussians and camera poses in $SE(3)$ Bézier space using the Adam optimizer. For Bézier, we use 9 control points. The learning rate for Gaussians follows the original 3DGS Kerbl et al. (2023a), while for camera poses, it is set to $1 \times 10^{-3}$. We set the number of virtual camera poses ($n$ in Eq. 2) to 15, ensuring a balance between performance and efficiency. We use 13 event bins for event-based deblurring (EDI). All experiments are conducted on an NVIDIA RTX 4090 GPU with a data factor of 2, using 7k iterations for all experiments and comparisons.

## D.2   Quantitative Results

**Table 7: Quantitative comparisons for deblurring on the Synthetic Dataset.** The table is organized into three groups: (1) Methods using only motion-blurred images (*w/o Events*), (2) Methods using event data as additional input (*w/ Events*), and (3) Methods requiring sharp images for SfM initialization (*w/ Sharp Supervision*).

| Scene | Metric | w/o Events | | | w/ Events | | | | w/ Sharp Supervision(*) | |
|---|---|---|---|---|---|---|---|---|---|---|
| | | MPRNet | Restormer | GeMS (Ours) | EDI+3DGS | E2NeRF | EBAD-NeRF | GeMS-E (Ours) | ExBluRF* | BAD-Gaussians* |
| Bench | PSNR↑ | 25.35 | 26.39 | 30.43 | 26.23 | 26.85 | 28.73 | 34.23 | 32.58 | 33.06 |
| | SSIM↑ | 0.678 | 0.720 | 0.855 | 0.761 | 0.749 | 0.839 | 0.933 | 0.873 | 0.910 |
| | LPIPS↓ | 0.425 | 0.356 | 0.098 | 0.219 | 0.270 | 0.156 | 0.045 | 0.123 | 0.034 |
| Camellia | PSNR↑ | 24.84 | 25.14 | 29.45 | 21.43 | 24.78 | 24.33 | 30.12 | 28.29 | 29.45 |
| | SSIM↑ | 0.669 | 0.690 | 0.848 | 0.667 | 0.702 | 0.753 | 0.890 | 0.744 | 0.834 |
| | LPIPS↓ | 0.395 | 0.351 | 0.110 | 0.280 | 0.232 | 0.175 | 0.092 | 0.318 | 0.086 |
| Dragon | PSNR↑ | 29.96 | 28.37 | 33.09 | 32.51 | 30.36 | 34.37 | 37.48 | 33.52 | 37.69 |
| | SSIM↑ | 0.731 | 0.704 | 0.837 | 0.808 | 0.752 | 0.873 | 0.932 | 0.831 | 0.938 |
| | LPIPS↓ | 0.454 | 0.465 | 0.150 | 0.241 | 0.268 | 0.198 | 0.060 | 0.192 | 0.039 |
| Jars | PSNR↑ | 25.36 | 25.57 | 32.22 | 27.12 | 26.52 | 29.35 | 33.30 | 30.50 | 32.30 |
| | SSIM↑ | 0.680 | 0.687 | 0.897 | 0.770 | 0.744 | 0.858 | 0.917 | 0.843 | 0.878 |
| | LPIPS↓ | 0.406 | 0.371 | 0.093 | 0.232 | 0.241 | 0.172 | 0.076 | 0.161 | 0.067 |
| Jars2 | PSNR↑ | 24.33 | 26.43 | 28.74 | 25.23 | 23.74 | 27.68 | 30.01 | 31.67 | 29.39 |
| | SSIM↑ | 0.745 | 0.814 | 0.882 | 0.786 | 0.732 | 0.874 | 0.921 | 0.901 | 0.852 |
| | LPIPS↓ | 0.358 | 0.275 | 0.129 | 0.221 | 0.313 | 0.151 | 0.090 | 0.118 | 0.091 |
| Postbox | PSNR↑ | 25.89 | 26.52 | 27.48 | 24.96 | 26.90 | 27.24 | 32.22 | 31.58 | 29.87 |
| | SSIM↑ | 0.736 | 0.753 | 0.760 | 0.767 | 0.784 | 0.832 | 0.913 | 0.858 | 0.823 |
| | LPIPS↓ | 0.318 | 0.286 | 0.164 | 0.210 | 0.172 | 0.146 | 0.061 | 0.134 | 0.075 |
| Stone Lantern | PSNR↑ | 24.97 | 26.68 | 28.63 | 25.55 | 25.14 | 27.40 | 30.38 | 28.24 | 29.66 |
| | SSIM↑ | 0.785 | 0.831 | 0.864 | 0.795 | 0.765 | 0.838 | 0.916 | 0.813 | 0.891 |
| | LPIPS↓ | 0.342 | 0.280 | 0.148 | 0.231 | 0.255 | 0.228 | 0.107 | 0.243 | 0.099 |
| Sunflowers | PSNR↑ | 28.86 | 29.55 | 30.60 | 29.14 | 28.67 | 31.29 | 34.63 | 32.92 | 34.57 |
| | SSIM↑ | 0.837 | 0.847 | 0.873 | 0.862 | 0.840 | 0.907 | 0.947 | 0.900 | 0.947 |
| | LPIPS↓ | 0.242 | 0.206 | 0.146 | 0.156 | 0.179 | 0.112 | 0.062 | 0.122 | 0.061 |
| Average | PSNR↑ | 26.19 | 26.83 | 30.08 | 26.52 | 26.62 | 28.80 | 32.80 | 31.16 | 32.00 |
| | SSIM↑ | 0.733 | 0.756 | 0.852 | 0.777 | 0.758 | 0.847 | 0.921 | 0.845 | 0.884 |
| | LPIPS↓ | 0.368 | 0.324 | 0.130 | 0.224 | 0.241 | 0.167 | 0.074 | 0.176 | 0.069 |

## D.3   Ablations

**Component-wise Analysis.**

- **w/o MCMC + w/o EDI + w/ VGGSfM**: Here we directly send extremely motion-blurred images to VGGSfM for Structure-from-Motion (SfM), which initializes camera poses and a sparse point cloud. This initialization is then passed to the standard (vanilla) 3DGS deblurring pipeline without using MCMC-based optimization. This variant serves as a baseline that excludes both 3DGS-MCMC probabilistic sampling and event-based deblurring.

- **w/ MCMC + w/o EDI + w/ HLOC**: In this setting, we again use extremely motion-blurred images, but SfM is performed using HLOC instead of VGGSfM. The resulting initialization is refined

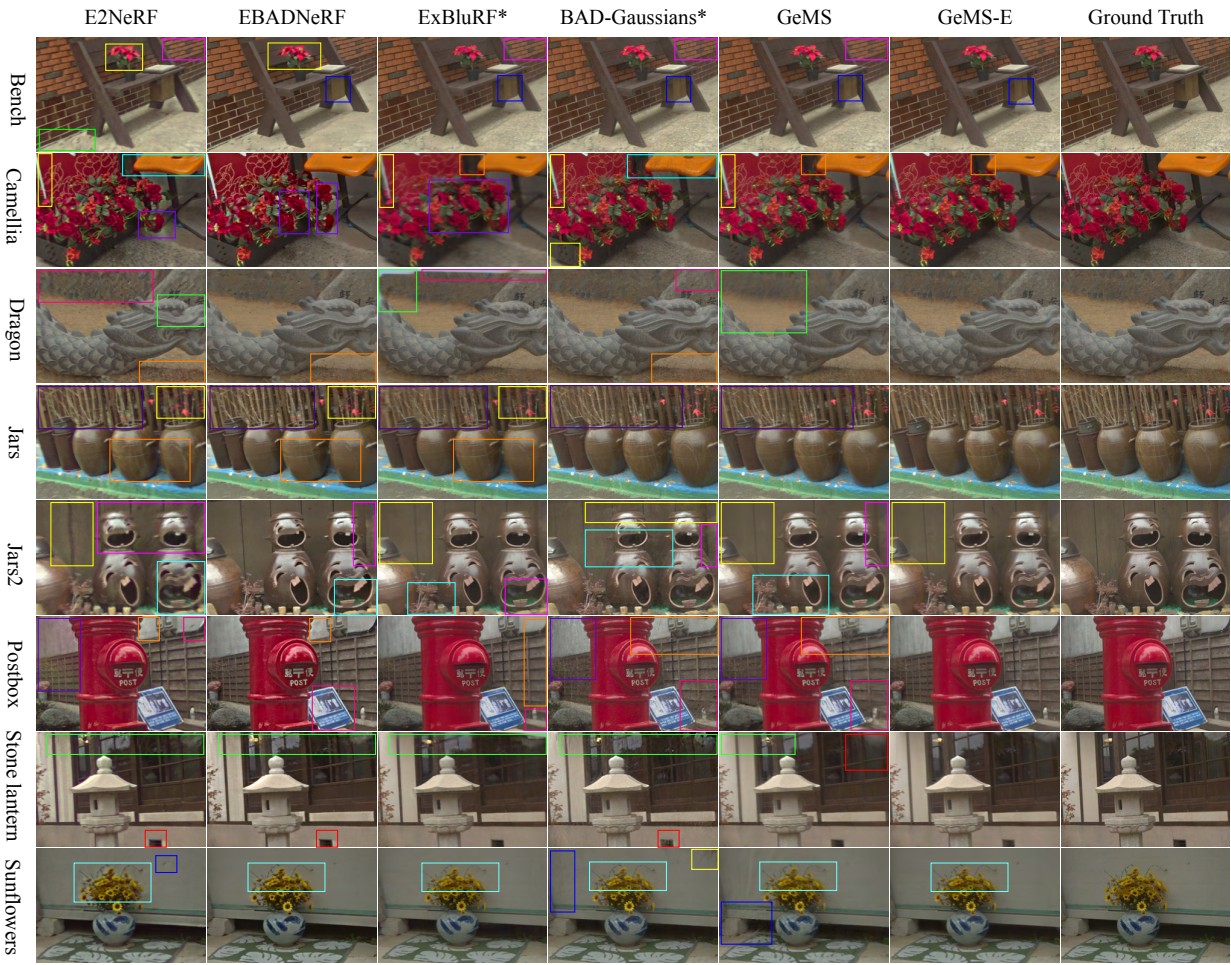

**Figure 14: Novel View Synthesis results on the Synthetic Dataset:** Our method effectively removes severe motion blur, reconstructing sharp novel views with high fidelity. Compared to existing approaches, it better preserves fine details and structural consistency while reducing color artifacts, demonstrating robustness under extreme blur conditions. Note that ExBluRF* and BAD-Gaussians* rely on pose and point cloud initializations from sharp images.

**Table 8:** MCMC robustness across blur point cloud initializations.

| Blur Level | w/o MCMC | | | w/ MCMC | | |
|---|---|---|---|---|---|---|
| | PSNR ↑ | SSIM ↑ | LPIPS ↓ | PSNR ↑ | SSIM ↑ | LPIPS ↓ |
| Sharp | 33.95 | 0.946 | 0.030 | **36.69** | **0.959** | **0.021** |
| 3 | 33.31 | 0.941 | 0.031 | **36.55** | **0.957** | **0.019** |
| 5 | 32.93 | 0.933 | 0.033 | **35.99** | **0.953** | **0.022** |
| 7 | 31.37 | 0.910 | 0.042 | **36.43** | **0.959** | **0.019** |
| 9 | 31.30 | 0.899 | 0.044 | **36.23** | **0.956** | **0.019** |
| 11 | 29.46 | 0.859 | 0.058 | **36.15** | **0.955** | **0.019** |

**Table 9:** MCMC robustness for deblurring across blur levels.

| Blur Level | w/o MCMC | | | w/ MCMC | | |
|---|---|---|---|---|---|---|
| | PSNR ↑ | SSIM ↑ | LPIPS ↓ | PSNR ↑ | SSIM ↑ | LPIPS ↓ |
| Sharp | 40.92 | 0.974 | 0.017 | **41.42** | **0.975** | **0.016** |
| 3 | 36.49 | 0.943 | 0.044 | **37.35** | **0.949** | **0.038** |
| 5 | 35.17 | 0.924 | 0.049 | **36.70** | **0.942** | **0.032** |
| 7 | 34.69 | 0.919 | 0.047 | **35.33** | **0.928** | **0.037** |
| 9 | 33.63 | 0.901 | 0.056 | **35.09** | **0.924** | **0.035** |
| 11 | 31.57 | 0.858 | 0.079 | **32.22** | **0.897** | **0.093** |

using MCMC-based optimization inside the 3DGS pipeline. Event-based deblurring is not applied. This configuration allows us to test an alternative SfM approach under blur.

- **GeMS (w/ MCMC + w/o EDI + w/ VGGSfM)**: This is our GeMS method, where we use VGGSfM for SfM initialization from motion-blurred images and apply MCMC-based optimization within the 3DGS framework. This variant highlights the impact of integrating VGGSfM with MCMC to handle blurry image sequences effectively.

- **w/o MCMC + w/ EDI + w/ VGGSfM**: In this case, we first deblur the motion-blurred images using the Event-based Deblurring Inference (EDI) technique. These deblurred images are then processed by VGGSfM for SfM, and the resulting initialization is fed into the standard 3DGS deblurring pipeline without MCMC refinement. This allows us to evaluate the benefits of EDI in isolation from MCMC probabilistic optimization.

- **w/ MCMC + w/ EDI + w/ COLMAP**: Here we first apply EDI to deblur the input images. SfM is then performed using COLMAP instead of VGGSfM, and MCMC-based optimization is used during 3DGS training. This configuration tests the interaction between EDI, MCMC, and COLMAP.

- **GeMS-E (w/ MCMC + w/ EDI + w/ VGGSfM)**: This is our full pipeline. We first apply EDI to deblur the input images, then use VGGSfM for SfM initialization, and finally perform MCMC-based joint optimization during 3DGS training. This variant combines all proposed components and is expected to offer the most robust performance under extreme motion blur.

**Table 10: Ablation study for deblurring on the Synthetic Dataset.** We assess the contribution of VGGSfM, MCMC, and EDI to deblurring performance. Each component progressively enhances PSNR, SSIM, and LPIPS, with their combined effect leading to the most accurate and visually consistent reconstructions.

| Method | Metric | Bench | Camellia | Dragon | Jars | Jars2 | Postbox | Stone L. | Sunflowers | Average |
|---|---|---|---|---|---|---|---|---|---|---|
| w/o MCMC + w/o EDI + w/ VGGSfM | PSNR↑ | 30.78 | 28.54 | 34.21 | 31.57 | 28.73 | 26.63 | 27.98 | 28.39 | 29.60 |
| | SSIM↑ | 0.853 | 0.801 | 0.857 | 0.858 | 0.841 | 0.694 | 0.846 | 0.823 | 0.822 |
| | LPIPS↓ | 0.080 | 0.101 | 0.083 | 0.079 | 0.096 | 0.176 | 0.137 | 0.234 | 0.123 |
| w/ MCMC + w/o EDI + w/ HLOC | PSNR↑ | 30.65 | 28.32 | 32.32 | 28.82 | 28.50 | 25.76 | 26.34 | 32.72 | 29.18 |
| | SSIM↑ | 0.838 | 0.774 | 0.807 | 0.755 | 0.831 | 0.664 | 0.770 | 0.909 | 0.794 |
| | LPIPS↓ | 0.085 | 0.123 | 0.110 | 0.136 | 0.097 | 0.186 | 0.186 | 0.073 | 0.124 |
| **GeMS (w/ MCMC + w/o EDI + w/ VGGSfM)** | PSNR↑ | 30.43 | 29.45 | 33.09 | 32.22 | 28.74 | 27.48 | 28.63 | 30.60 | 30.08 |
| | SSIM↑ | 0.855 | 0.848 | 0.837 | 0.897 | 0.882 | 0.760 | 0.864 | 0.873 | 0.852 |
| | LPIPS↓ | 0.098 | 0.110 | 0.150 | 0.093 | 0.129 | 0.164 | 0.148 | 0.146 | 0.130 |
| w/o MCMC + w/ EDI + w/ VGGSfM | PSNR↑ | 33.06 | 29.45 | 37.69 | 32.30 | 29.39 | 29.87 | 29.66 | 34.57 | 32.00 |
| | SSIM↑ | 0.910 | 0.834 | 0.938 | 0.878 | 0.852 | 0.823 | 0.891 | 0.947 | 0.884 |
| | LPIPS↓ | 0.034 | 0.086 | 0.039 | 0.067 | 0.091 | 0.075 | 0.099 | 0.061 | 0.069 |
| w/ MCMC + w/ EDI + w/ COLMAP | PSNR↑ | 32.93 | 30.68 | 35.56 | 32.14 | 29.64 | 31.39 | 29.34 | 34.31 | 32.00 |
| | SSIM↑ | 0.924 | 0.890 | 0.879 | 0.894 | 0.898 | 0.895 | 0.872 | 0.941 | 0.899 |
| | LPIPS↓ | 0.059 | 0.092 | 0.173 | 0.115 | 0.118 | 0.088 | 0.182 | 0.075 | 0.113 |
| **GeMS-E (w/ MCMC + w/ EDI + w/ VGGSfM)** | PSNR↑ | 34.23 | 30.12 | 37.48 | 33.30 | 30.01 | 32.22 | 30.38 | 34.63 | 32.80 |
| | SSIM↑ | 0.933 | 0.890 | 0.932 | 0.917 | 0.921 | 0.913 | 0.916 | 0.947 | 0.921 |
| | LPIPS↓ | 0.045 | 0.092 | 0.060 | 0.076 | 0.090 | 0.061 | 0.107 | 0.062 | 0.074 |

### D.3.1 Comparative Robustness of GeMS vs BAD-Gaussians under Motion Blur

We conducted a systematic comparison of GeMS and BAD-Gaussians across increasing levels of motion blur to assess their robustness and practical applicability. For this evaluation, COLMAP was used to provide SfM initializations for BAD-Gaussians, while VGGSfM was used for GeMS. COLMAP is able to generate valid initializations only up to moderate blur (up to 7-frame averaging); at higher blur levels, its feature matching pipeline fails, as summarized in Table 2. In contrast, VGGSfM consistently produces robust initializations even at extreme blur levels (up to 11-frame averaging).

To compare downstream performance, we provided each method with its respective initialization and evaluated reconstruction quality across all blur settings. As shown in Figure 15 and Table 11, GeMS consistently outperforms BAD-Gaussians at all comparable blur levels. Notably, as blur severity increases, BAD-Gaussians' performance drops sharply and becomes inapplicable once COLMAP fails, while GeMS remains robust and continues to deliver high-quality reconstructions. This analysis highlights GeMS's ability to handle severe motion blur scenarios where traditional pipelines are no longer viable.

**Table 11:** Deblurring performance comparison between GeMS and BAD-Gaussians at various blur levels.

| Blur Level | GeMS | | | BAD-Gaussians | | |
| --- | --- | --- | --- | --- | --- | --- |
| | PSNR | SSIM | LPIPS | PSNR | SSIM | LPIPS |
| Sharp | 41.42 | 0.975 | 0.016 | 41.02 | 0.974 | 0.017 |
| 3 | 37.35 | 0.949 | 0.038 | 37.14 | 0.946 | 0.036 |
| 5 | 36.70 | 0.942 | 0.032 | 33.72 | 0.901 | 0.069 |
| 7 | 35.33 | 0.928 | 0.037 | 30.99 | 0.842 | 0.111 |
| 9 | 35.09 | 0.924 | 0.035 | x | x | x |
| 11 | 32.22 | 0.897 | 0.093 | x | x | x |

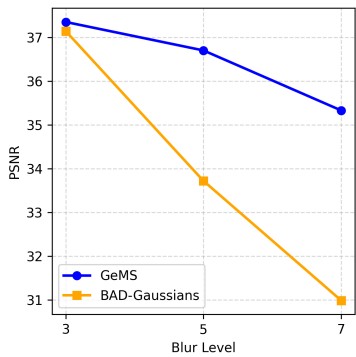

**Figure 15:** PSNR comparison between GeMS and BAD-Gaussians across various blur levels.

# E  Qualitative Results

## E.1  Synthetic Dataset(View-1)

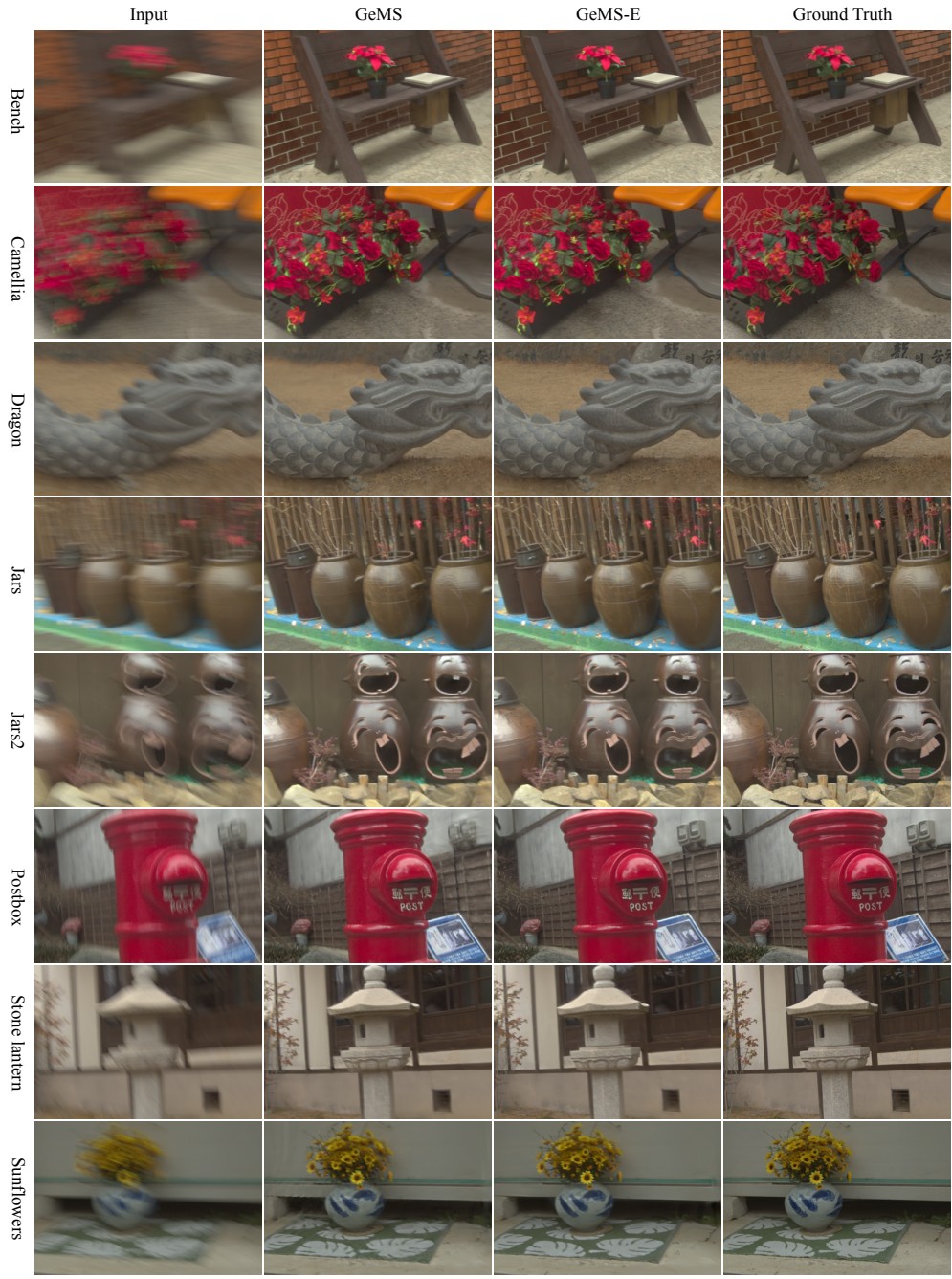

Figure 16: Our Results on Synthetic Dataset

## E.2   Synthetic Dataset(View-2)

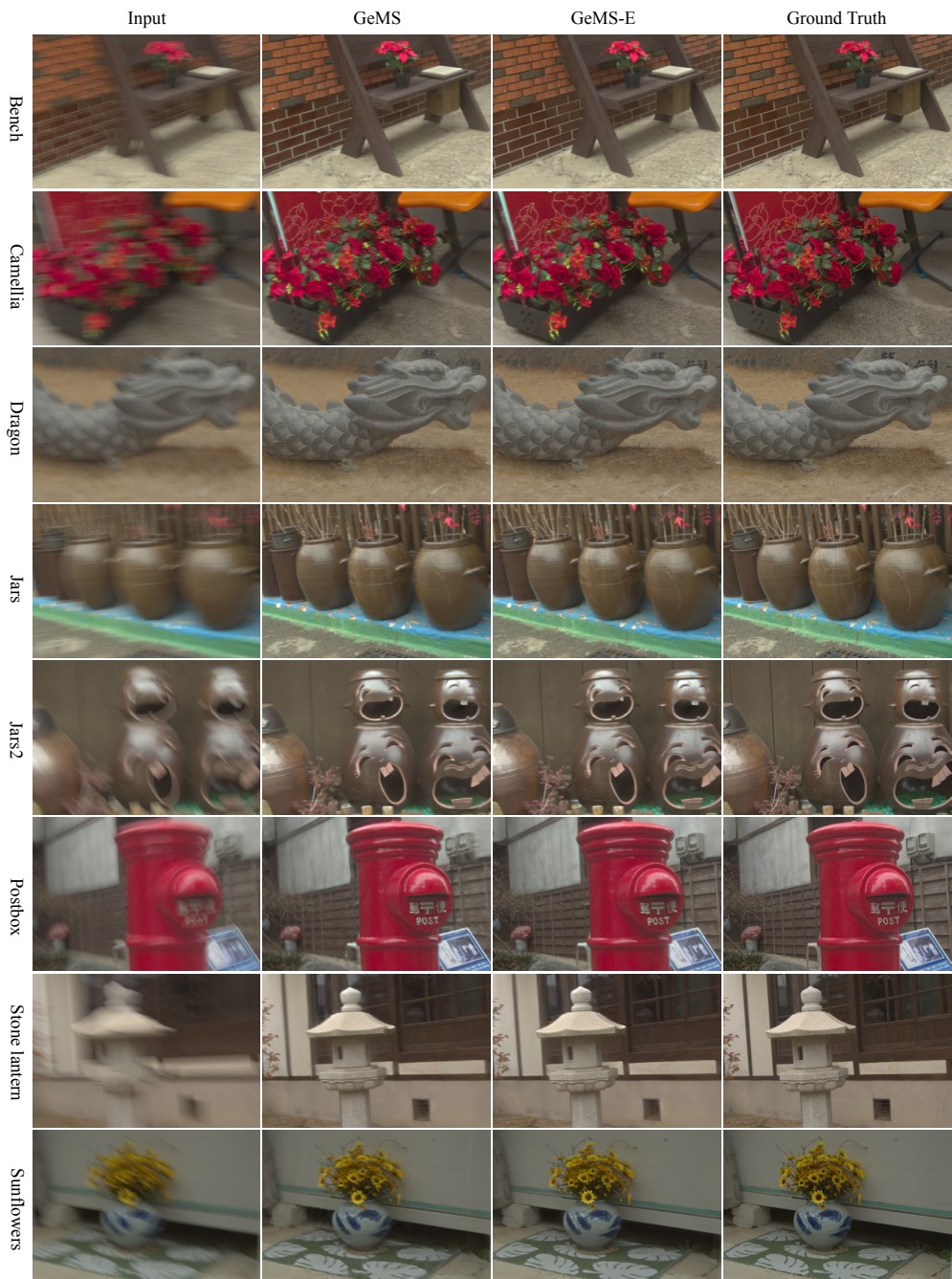

Figure 17: Our Results on Synthetic Dataset

## E.3    Real Dataset(View-1)

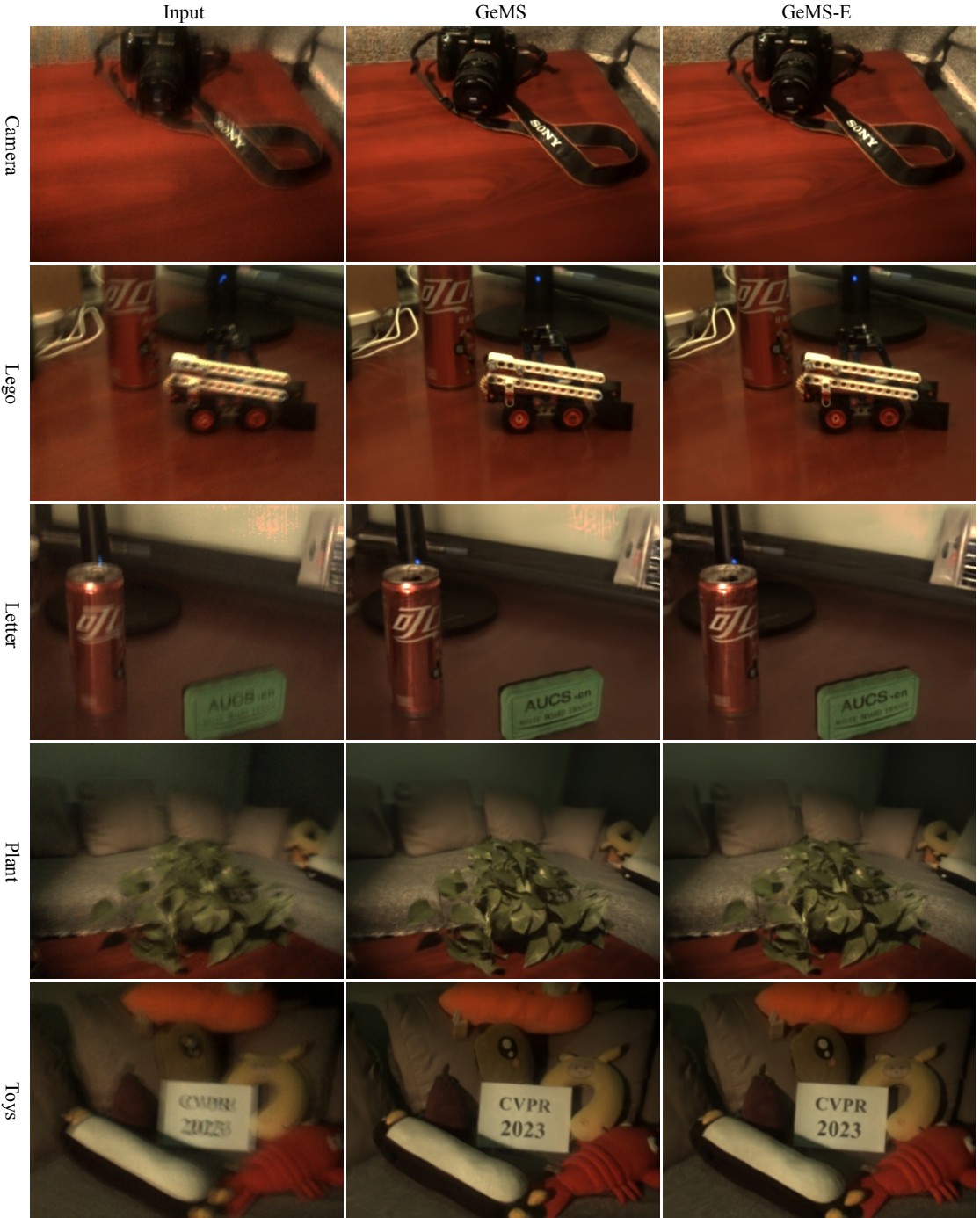

Figure 18: Our Results on Real Dataset

## E.4 Real Dataset(View-2)

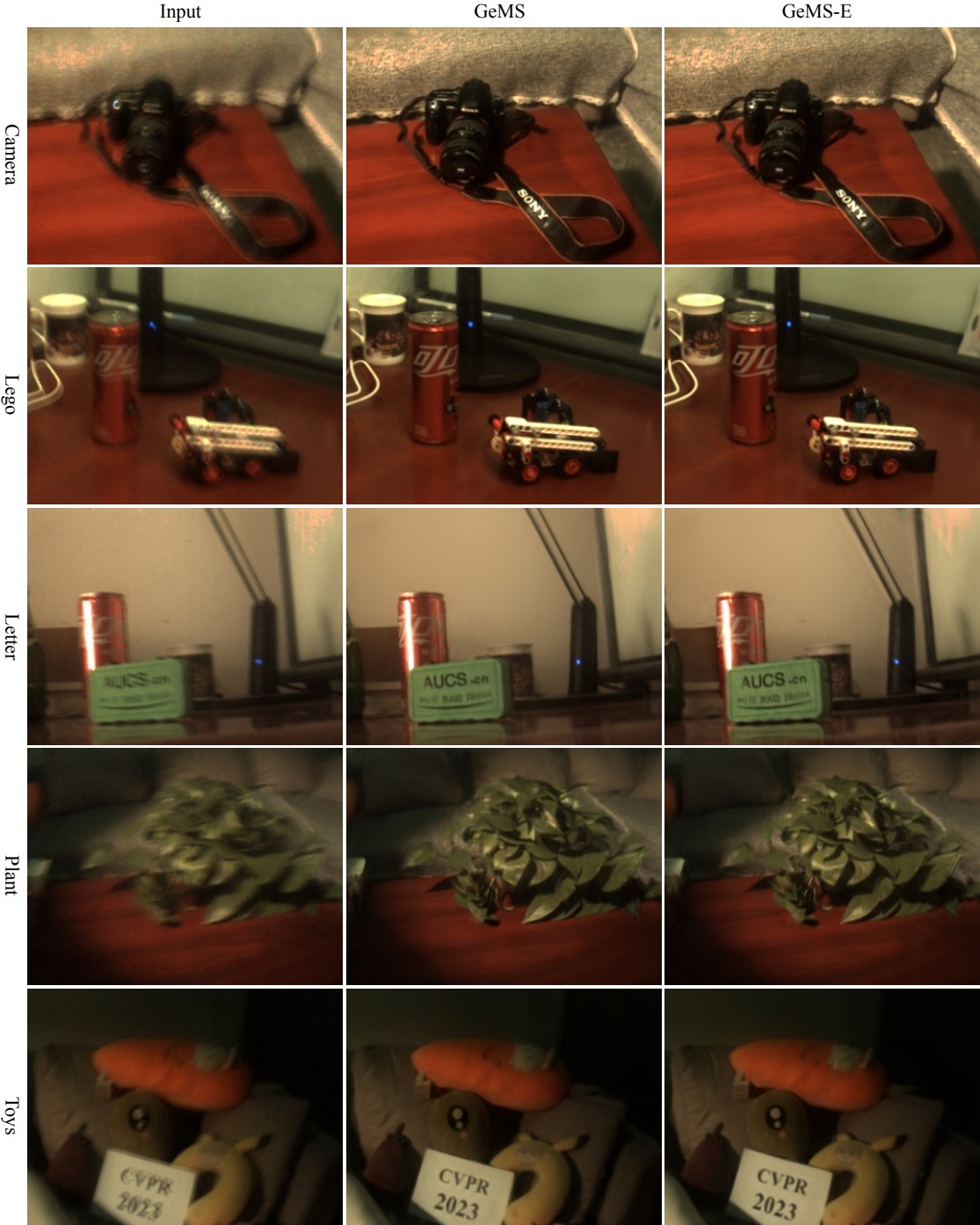

Figure 19: **Our Results on Real Dataset**

