# OpenReview forum: "GeMS: Efficient Gaussian Splatting for Extreme Motion Blur"
_TMLR — Rejected by TMLR_

### Review · Reviewer_ExwN · 2025-05-31

**Summary Of Contributions:**

GeMS focuses on reconstructing a 3D scene with Gaussian Splatting utilizing severely motion-blurred images. It utilizes VGGSfM to estimate camera poses from severely motion-blurred images, and incorporates the 3DGS-MCC method to initialize and densify the scene. Then it jointly optimizes camera poses and Gaussians using BAD-Gaussian. When event data is available, GeMS-E is introduced to improve the reconstruction quality by incorporating the EDI model to deblur images. Experiments on synthetic and real-world datasets show that GeMS and GeMS-E achieve SOTA performance in deblurring and NVS tasks.

**Audience:**

Yes

**Broader Impact Concerns:**

I don't think there are any broader impact concerns.

**Claims And Evidence:**

Yes

**Requested Changes:**

1. Table 1, I understand that ExBluRF and BAD-Gaussians are excluded from the main rankings for fairness, still, the highlight remarks are misleading. Meanwhile, GeMS-R uses event data and some other methods do not; that's also not a fair comparison. I think this table should be better organized.
2. The Figures show deblur results. How are the novel view synthesis results?

**Strengths And Weaknesses:**

Strengths:
1. The method efficiently utilizes useful components, and to be honest, I really like that the authors do not overclaim their contributions and say everything as it is.
2. This task makes sense, I agree that this work is more practical than works that use sharp images and COLMAP to obtain camera poses.
3. The results are good, both quantitatively and qualitatively. All modules are properly evaluated by ablation studies.

Weaknesses:
1. I believe this paper is more engineering-oriented than research-focused, as it combines existing works that achieve good results.
2. References should be better organized, e.g., BAD-Gaussians is referenced twice.
3. The experiment sections should be better organized. See requested changes.

---

> ### Author Response · Authors · 2025-06-26
> **To Reviewer ExwN**
>
> We sincerely thank the reviewer for taking the time to carefully read our manuscript and for providing thoughtful and constructive feedback. Your insights have been invaluable in helping us improve the clarity, rigor, and overall quality of our work. We have carefully addressed each point below and made the corresponding revisions to the manuscript. Our responses are organized point-by-point for clarity.
>
> ---
>
> ### R-ExwN.1
> > References should be better organized, e.g., BAD-Gaussians is referenced twice.
>
> **Response**
> We have carefully reviewed and corrected our bibliography to remove duplicate entries, including the BAD-Gaussians reference. All references are now consistently cited and organized according to best practices, ensuring clarity and eliminating redundancy in the reference list.
>
> ---
>
> ### R-ExwN.2
> > I understand that ExBluRF and BAD-Gaussians are excluded from the main rankings for fairness, still, the highlight remarks are misleading. Meanwhile, GeMS-R uses event data and some other methods do not; that's also not a fair comparison. I think this table should be better organized.
>
> **Response**
> Thank you for this suggestion. We have reorganized **Table 1** into three clearly separated groups for fair comparison:
>
> 1. Methods using only motion-blurred images (*w/o Events*),
> 2. Methods using event data as additional input (*w/ Events*), and
> 3. Methods requiring sharp images for SfM initialization (*w/ Sharp Supervision*).
>
> Metric rankings and highlight colors are now reported separately within **Groups 1 and 2** to avoid misleading cross-group comparisons. Group 3 methods such as **ExBluRF\*** and **BAD-Gaussians\*** are included *only for reference* and are not ranked, as they rely on sharp images and are not directly comparable. This reorganization ensures a more accurate and fair evaluation of each method in its appropriate context.
>
> ---
>
> ### R-ExwN.3
> > The Figures show deblur results. How are the novel view synthesis results?
>
> **Response**
> To address this concern, we specifically included a figure showcasing **novel view synthesis results** in the appendix due to space limitations in the main paper. Please refer to **Figure 14**. This figure provides a clear visual comparison of our method’s performance on novel view synthesis, in addition to the deblurring results. It demonstrates that **GeMS-E produces high-fidelity and consistent novel views** even in the presence of severe motion blur, further validating the effectiveness of our pipeline for both image reconstruction and view synthesis.

---

### Review · Reviewer_kTey · 2025-06-12

**Summary Of Contributions:**

This paper introduces a new system to reconstruct high-quality 3DGS from motion-blurred images. The framework consists of three parts 1) using VGGSfM instead of COLMAP, 2) using 3DGS-MCMC, and 3) using an event camera to deblur the images. The final resulting system produces more reasonable results than previous deblur-GS methods that use clean images for pose estimation.

**Audience:**

Yes

**Claims And Evidence:**

No

**Requested Changes:**

Since the paper does not bring anything new, I would regard this as a system paper, which should contain a more detailed discussion about the application scope of each component.

1. For the VGGSfM, 1) why can it handle severely blurry images? Because I think the pose is somehow ambiguous about which pose is accurate for this image. 2) To what extent can VGGSfM handle motion blur? There should be a systematic analysis of it.
2. Similarly, for 3DGS-MCMC, is 3DGS-MCMC especially effective in terms of handling blurs? Because the original 3DGS-MCMC paper already shows improvements over the original 3DGS. How and why it even works for blurry images is not analyzed in detail.
3. Why did we choose this algorithm for event image-based deblurring? A straightforward way is to just use the event images to deblur the images and then estimate the camera poses. I assume that the baseline here only applies deblur in the 3DGS training instead of in the pose estimation. Is there any other deburring algorithm applicable here, and how effective is it?

**Strengths And Weaknesses:**

Strength
1. The paper is well-written.
2. Addressing the pose estimation problem is realistic and could make the system more practical.

Weakness
1. This paper does not propose anything new. It just combines the VGGSfM, 3DGS-MCMC, and event-based deblur algorithms.
2. The experimental analyses are too simple, which just compare with baselines and ablate some components.

---

> ### Author Response · Authors · 2025-06-26
> **To Reviewer kTey**
>
> ### R-kTey.1
> > Since the paper does not bring anything new, I would regard this as a system paper, which should contain a more detailed discussion about the application scope of each component.
>
> #### R-kTey.1a
> > For the VGGSfM, why can it handle severely blurry images? Because I think the pose is somehow ambiguous about which pose is accurate for this image. To what extent can VGGSfM handle motion blur? There should be a systematic analysis of it.
>
> **Response**
> Classical frameworks like COLMAP solve structure-from-motion incrementally through keypoint detection, matching, and triangulation, which makes them highly sensitive to blur since texture loss disrupts feature matching at the outset. In contrast, VGGSfM employs deep 2D point tracking and a fully differentiable pipeline, extracting reliable pixel-accurate tracks even when texture is severely degraded and directly predicting global correspondences.
>
> To systematically validate VGGSfM’s robustness to motion blur, we perform an experiment where we synthesized blurred images by averaging multiple consecutive sharp frames, simulating increasing levels of motion blur. In this setup, both VGGSfM and COLMAP were evaluated on their ability to register images and estimate accurate camera poses and point clouds as blur severity increased. Our results show that while COLMAP fails to register images beyond moderate blur, VGGSfM consistently succeeds across all blur levels, producing stable poses and dense point clouds as detailed in Table 2, Table 3, and Figure 7. This systematic evaluation demonstrates that VGGSfM’s design enables robust and reliable 3D reconstruction from severely blurred images.
>
> ---
>
> #### R-kTey.1b
> > Similarly, for 3DGS-MCMC, is 3DGS-MCMC especially effective in terms of handling blurs? Because the original 3DGS-MCMC paper already shows improvements over the original 3DGS. How and why it even works for blurry images is not analyzed in detail.
>
> **Response**
> 3DGS-MCMC is uniquely effective for motion blur due to its probabilistic framework, which we rigorously validated through two experiments (Section 3.4.1 *3DGS-MCMC Robustness*). In the first experiment, we evaluated robustness to blur-corrupted initializations by generating point clouds from images with progressive motion blur (blur-3 to blur-11) using VGGSfM. When initializing vanilla 3DGS and 3DGS-MCMC with these degraded inputs (loss computed against sharp ground truth), standard 3DGS suffers significant performance degradation as blur increases, whereas 3DGS-MCMC maintains stable reconstruction quality across all blur levels.
>
> This quantitatively validates that the probabilistic sampling framework of MCMC is inherently robust to initialization quality, and specifically effective for blur-corrupted point cloud initializations.
>
> In the second experiment, we assessed deblurring performance within the GeMS framework by comparing variants with and without MCMC, both optimized against original blurry images. GeMS with MCMC consistently outperformed the non-MCMC variant across blur levels. Also, qualitative ablations (Figure 10 & 11) reveal that the non-MCMC variant exhibits noticeable artifacts and structural inconsistencies in extreme motion blur scenarios, highlighting the practical value of MCMC for reliable scene recovery under challenging blur conditions.
>
> ---

---

> > ### Author Response · Authors · 2025-06-26
> > **To Reviewer kTey**
> >
> > #### R-kTey.1c
> > > Why did we choose this algorithm for event image-based deblurring? A straightforward way is to just use the event images to deblur the images and then estimate the camera poses. I assume that the baseline here only applies deblur in the 3DGS training instead of in the pose estimation. Is there any other deblurring algorithm applicable here, and how effective is it?
> >
> > **Response**
> > We chose the **Event-based Double Integral (EDI)** algorithm for event image-based deblurring because it is physically grounded in event camera principles and reliably reconstructs sharp images from severely blurred inputs. In our GeMS-E framework, these deblurred images are used exclusively for initializing the SfM stage (VGGSfM), providing accurate camera poses and sparse geometry that would otherwise be unattainable from blurred images alone.
> >
> > Importantly, the subsequent reconstruction process is guided not by these deblurred images, but by the original blurry images using our physics-based blur formation model, with the photometric loss computed against the observed blur. This selective integration ensures that event data enhances initialization without compromising the physical fidelity of blur modeling.
> >
> > In contrast, a straightforward approach—simply deblurring the images with EDI and then using them for standard 3DGS reconstruction—yields subpar results, as it cannot correct residual deblurring artifacts or resolve ambiguities between pose and scene geometry, as shown in Table 1 (EDI+3DGS, column 4).
> >
> > We also evaluated non-event-based deblurring algorithms such as **Restormer** and **MPRNet**. While these methods can improve image sharpness to some extent, they do not leverage the high temporal resolution and motion information provided by event data. As a result, they are less effective in restoring details and structure from severely motion-blurred inputs, and their use as a preprocessing step for 3DGS leads to limited improvements in reconstruction quality.
> >
> > In addition, we considered event-driven baselines like **E2NeRF** and **EBAD-NeRF**. Although these methods are designed to incorporate event information, they still struggle under conditions of extreme blur or sparse geometry. E2NeRF and EBAD-NeRF lack the probabilistic refinement present in GeMS-E, making them less robust in challenging scenarios.
> >
> > Our integrated approach—which combines event-based deblurring, robust initialization, probabilistic scene modeling, and joint pose-geometry optimization—consistently outperforms these alternatives in both synthetic and real-world settings.
> >
> > ---
> >
> > We sincerely thank the reviewer for taking the time to carefully read our manuscript and for providing thoughtful and constructive feedback. Your insights have been invaluable in helping us improve the clarity, rigor, and overall quality of our work. We have carefully addressed each point below and made the corresponding revisions to the manuscript. Our responses are organized point-by-point for clarity.

---

### Review · Reviewer_vp37 · 2025-06-12

**Summary Of Contributions:**

The paper proposes GeMS (-E), a framework designed to address the challenge of handling severely motion-blurred images within the 3D Gaussian Splatting paradigm. The approach integrates several components: a deep-learning-based VGGsfM in place of COLMAP, an MCMC-based Gaussian splatting method, a joint optimization procedure, and the use of additional event data. While the empirical results seem promising, from the perspective of a novice reader, the paper seems hard to understand due to unclear explanations and presentation. It is difficult to understand the motivation behind this particular combination of components; why each part is necessary, what specific role it plays (why the previous method is problematic), and what makes the integration of these elements novel. Overall, I believe the paper requires substantial revision to clarify its contributions and justify the design choices.

**Audience:**

Yes

**Broader Impact Concerns:**

I included them all in the weakness section.

**Claims And Evidence:**

No

**Requested Changes:**

I wrote them all in weaknessses parts, and I believe that improving the clarity of the paper should be the first priority. Once it's revised for better readability, I will be able to properly evaluate its contributions. Although I think the paper requires major revision (but considering I am a novice reader), I would also like to see the other reviewers' opinions

**Strengths And Weaknesses:**

Strength:
The paper appears to tackle an important and challenging problem (targeting severe motion blur under  3D Gaussian Splatting). The proposed framework, GeMS (-E), shows promising empirical results.

Weaknesses:
1) **Clarity and Presentation**
The most critical issue is the clarity of the paper. In my opinion, academic papers should be written in a self-contained manner, accessible even to non-expert or novice readers. However, this paper is difficult to follow as a novice reader

- The abstract is dense and overloaded with unexplained abbreviations (e.g., SfM, MCMC) and lists too many methods without proper explanation. It means that the abstract lacks sufficient generality and clarity. Moreover, Phrases like "COLMAP fails" are vague what specifically does "fail" mean in this context? These details should either be clarified or deferred to later sections such as the related work.

- Introduction & Related Work: These sections are not well separated. The introduction contains a mixture of background and literature review (Many parts of the introduction seem like just simply listing previous literature without specific and refined information), making it hard to follow the paper’s central claim. I strongly recommend structuring the introduction to clearly state; 1) the motivation of the problem, 2) the necessities of each proposed component, and 3) an overview of how the proposed method addresses the problem.

- For example, authors should explain 1) Why is VGGsfM particularly suitable for initialization in motion-blurred scenarios, and in what sense is it a viable "COLMAP-free" solution? 2) Why is a probabilistic formulation appropriate or beneficial for coarse/sparse motion-blur settings? 3) What is the role of joint optimization, and why is it necessary? 4) How does the proposed combination of modules produce synergy, rather than just functioning as an ensemble of known components?

- In the method section: 1) Is VGGsfM originally designed for 2D? If so, how is it adapted for use in 3D Gaussian Splatting? 2) The probabilistic interpretation of 3DGS-MCMC is not clearly described. What has been reinterpreted probabilistically, and why is this reformulation helpful? 3) The purpose and implementation of the joint optimization step are not well motivated or explained.

- Several terms and concepts are introduced without definition for properly understanding the method. 1) What is SE(3) in this context 2) What D-SSIM term represent? 3) How is the L1 loss formulated? 4) What is the exact role of "event data" and how is it integrated?

- The presentation of Ablation Table 2 could be improved by clearly denoting each column header.

2. **Contributions.** While combining multiple existing techniques can be a valuable contribution, the paper must clearly justify: 1) Why each component is necessary for this specific problem, 2) What the novelty is in their combination, 3) How the combination is greater than the sum of its parts. However, the current state of the paper cannot explain the rationale behind the combination, and in fact, hard to understand why each component is needed.

---

> ### Author Response · Authors · 2025-06-26
> **To Reviewer vp37**
>
> First of all, we sincerely thank the reviewer for taking the time to carefully read our manuscript and for providing thoughtful and constructive feedback. Your insights have been invaluable in helping us improve the clarity, rigor, and overall quality of our work. We have carefully addressed each point below and made the corresponding revisions to the manuscript. Our responses are organized point-by-point for clarity.
>
> ### R-vp37.1
> > The abstract is dense and overloaded with unexplained abbreviations (e.g., SfM, MCMC) and lists too many methods without proper explanation. It means that the abstract lacks sufficient generality and clarity. Moreover, Phrases like "COLMAP fails" are vague what specifically does "fail" mean in this context? These details should either be clarified or deferred to later sections such as the related work.
>
> **Response**
> Thank you for highlighting these issues. Now we revise the abstract to improve clarity and accessibility: all abbreviations (such as SfM and MCMC) are now defined at first use. We also clarified what is meant by "COLMAP fails," specifying that COLMAP is unable to reliably estimate camera poses or point clouds in the presence of extreme motion blur.
>
> ---
>
> ### R-vp37.2
> > Introduction & Related Work: These sections are not well separated. The introduction contains a mixture of background and literature review (Many parts of the introduction seem like just simply listing previous literature without specific and refined information), making it hard to follow the paper’s central claim. I strongly recommend structuring the introduction to clearly state; (1) the motivation of the problem, (2) the necessities of each proposed component, and (3) an overview of how the proposed method addresses the problem.
>
> **Response**
> Thank you for this valuable feedback. Now we revise the manuscript to clearly separate the introduction and related work sections. The introduction now directly states the motivation for the problem, outlines the necessity of each proposed component, and provides a concise overview of how our method addresses the challenges of 3D reconstruction from motion-blurred images. The related work section now focuses on summarizing and contextualizing prior literature, avoiding excessive listing in the introduction. Due to the lack of space in main paper, related works section is presented in Appendix A.
>
> ---

---

> > ### Author Response · Authors · 2025-06-26
> > **To Reviewer vp37**
> >
> > ### R-vp37.3a
> > > Why is VGGsfM particularly suitable for initialization in motion-blurred scenarios, and in what sense is it a viable "COLMAP-free" solution?
> >
> > **Response**
> > COLMAP, as a traditional structure-from-motion (SfM) pipeline, relies heavily on detecting and matching keypoints across images—a process that breaks down under severe motion blur, where image textures are smeared and reliable features are lost. This makes COLMAP unsuitable for scenarios with extreme blur, as it struggles to initialize camera poses and point clouds, please refer to Table 2. VGGSfM, in contrast, is a fully differentiable, deep learning-based SfM framework that leverages robust 2D point tracking and global optimization. It extracts reliable pixel-accurate tracks without relying on pairwise feature matches, and estimates all camera poses simultaneously using both image and track features. The pipeline includes a differentiable bundle adjustment layer that jointly refines camera parameters and 3D points, making it especially effective in challenging extreme motion blur conditions where COLMAP fails (See Table 2). In our work, “COLMAP-free” means we do not use COLMAP at any stage; instead, we use VGGSfM for initialization, which has proven to work reliably even with extremely motion-blurred images, providing a viable and robust alternative for SfM in these scenarios.
> >
> > ---
> >
> > ### R-vp37.3b
> > > Why is a probabilistic formulation appropriate or beneficial for coarse/sparse motion-blur settings?
> >
> > **Response**
> > 3DGS-MCMC is robust to blurry or sparse point cloud initializations because its probabilistic formulation allows the reconstruction to adaptively handle uncertainty and incomplete data, which are common in motion-blur scenarios. By modeling 3D Gaussians as samples from a probability distribution, 3DGS-MCMC can dynamically refine and densify the scene without relying on heuristic rules, making it effective even when initial point clouds are noisy or lack detail. This flexibility leads to more stable and accurate reconstructions under severe blur, as demonstrated in our experiments, please see Figures 8 and 9 where 3DGS-MCMC consistently outperforms standard 3DGS as the quality of the initialization degrades.
> >
> > ---
> >
> > ### R-vp37.3c
> > > What is the role of joint optimization, and why is it necessary?
> >
> > **Response**
> > Joint optimization plays a critical role in our pipeline by enabling the simultaneous refinement of camera trajectories and 3D scene geometry, which is especially necessary in the context of extreme motion blur. At the initial stage from SfM, we have only one pose for each blurry image, which is not sufficient to represent the continuous motion trajectory that actually occurs during exposure. Since every blurry image is formed by integrating scene information along this motion path, accurately modeling and inverting the blur requires optimizing both the trajectory and the scene geometry together. By jointly optimizing these components, our method can recover the true dynamics of camera motion, and ensure that the reconstructed scene and the modeled blur are physically consistent with the observed images. This joint refinement is essential for achieving high-fidelity reconstruction in scenarios dominated by severe motion blur.
> >
> > ---
> >
> > ### R-vp37.3d
> > > How does the proposed combination of modules produce synergy, rather than just functioning as an ensemble of known components?
> >
> > **Response**
> > The proposed combination of modules in GeMS produces true synergy by integrating all components into a single, end-to-end differentiable pipeline, rather than simply stacking known techniques. After initialization with VGGSfM, probabilistic scene modeling with 3DGS-MCMC and joint pose–geometry optimization are tightly coupled, allowing gradients and error signals to flow across modules during optimization. This design enables each part of the system to compensate for the weaknesses of the others: for example, 3DGS-MCMC adaptively refines and densifies the often noisy or sparse outputs from VGGSfM, while joint optimization co-refines camera trajectories and scene geometry to ensure consistency with the observed data. Event-based deblurring is seamlessly incorporated to provide sharp priors only when needed, further strengthening the pipeline’s robustness. As a result, the system achieves a level of accuracy and stability in reconstructing 3D scenes from extremely motion-blurred images that cannot be matched by optimizing each component in isolation or by simply combining existing methods without such tight integration.
> >
> > ---

---

> > > ### Author Response · Authors · 2025-06-26
> > > **To Reviewer vp37**
> > >
> > > ### R-vp37.4a
> > > > Is VGGsfM originally designed for 2D? If so, how is it adapted for use in 3D Gaussian Splatting?
> > >
> > > **Response**
> > > VGGSfM is a deep learning-based structure-from-motion (SfM) pipeline that begins with 2D image sequences and uses deep 2D point tracking to establish reliable pixel-level correspondences across images. VGGSfM is explicitly designed to reconstruct both camera poses and a 3D point cloud from these 2D tracks using differentiable triangulation and bundle adjustment. In our pipeline, we use VGGSfM in the same way COLMAP is typically used: to obtain the initial camera parameters and a sparse 3D point cloud from the input images, even under severe motion blur. This 3D initialization is then directly used by our efficient 3D Gaussian Splatting (3DGS) framework.
> > >
> > > ---
> > >
> > > ### R-vp37.4b
> > > > The probabilistic interpretation of 3DGS-MCMC is not clearly described. What has been reinterpreted probabilistically, and why is this reformulation helpful?
> > >
> > > **Response**
> > > Thank you for pointing this out, now we make it more clear, please see Section 2.1.1. In 3DGS-MCMC, the set of 3D Gaussians is treated as samples from a probability distribution, rather than as fixed points determined by heuristics. This means that, instead of manually deciding where and how many Gaussians to place in the scene, the algorithm uses a probabilistic sampling process (specifically, Markov Chain Monte Carlo) to explore and adaptively fill in the scene geometry. This reformulation is helpful because it allows the method to automatically handle uncertainty, fill gaps, and avoid overfitting to noise—especially important when the initial data is sparse or noisy, as is common with motion-blurred images. As a result, the reconstruction process becomes more robust and stable, producing better results without relying on manual rules or perfect initializations, please refer Figure 8 and 9.
> > >
> > > ---
> > >
> > > ### R-vp37.4c
> > > > The purpose and implementation of the joint optimization step are not well motivated or explained.
> > >
> > > **Response**
> > > The joint optimization step is crucial because it directly models the physics of motion blur, where each blurry image is formed by integrating sharp scene content along the camera’s continuous trajectory during exposure. Initial poses from VGGSfM provide only a single viewpoint per image, which is not sufficient to capture the true motion path responsible for the observed blur. By jointly optimizing both the camera trajectories (modeled, for example, with Bézier curves) and the 3D Gaussian scene representation, our method can accurately invert the blur formation process and ensure that the reconstructed scene is physically consistent with the observed images. This co-adaptation allows the system to iteratively correct both trajectory and geometry errors, aligning the synthesized blur with the real blur in the input, and significantly reduces artifacts and misalignments. Our ablation studies further confirm that joint optimization with Bezier trajectory representation leads to much better results, please refer Figure 12.
> > >
> > > ---

---

> > > > ### Author Response · Authors · 2025-06-26
> > > > **To Reviewer vp37**
> > > >
> > > > ### R-vp37.5a
> > > > > What is SE(3) in this context?
> > > >
> > > > **Response**
> > > > SE(3) refers to the **Special Euclidean Group in 3D**, which represents rigid transformations combining 3D rotations and translations. In our method, SE(3) parameterizes camera trajectory poses (position and orientation) during optimization which is essential to model blur image formation process. For example, joint optimization adjusts SE(3) parameters to correct errors in initial camera trajectories under blur.
> > > >
> > > > ---
> > > >
> > > > ### R-vp37.5b
> > > > > What D-SSIM term represent?
> > > >
> > > > **Response**
> > > > D-SSIM (1 - SSIM) measures how different two images are in terms of structure, contrast, and luminance. We use it to evaluate the dissimilarity between reconstructed and ground-truth images, with lower D-SSIM indicating better perceptual quality.
> > > >
> > > > ---
> > > >
> > > > ### R-vp37.5c
> > > > > How is the L1 loss formulated?
> > > >
> > > > **Response**
> > > > The L1 loss computes the mean absolute difference between the reconstructed and ground-truth images. Given a reconstructed image `I_rec` and a ground-truth image `I_gt`, the L1 loss is defined as:
> > > >
> > > > **L1 = (1/N) × Σ |I_rec(i) − I_gt(i)|**
> > > >
> > > > where `N` is the total number of pixels, and the summation runs over all pixel indices `i`. This loss penalizes pixel-wise differences linearly, making it robust to outliers and suitable for preserving overall image structure.
> > > >
> > > > ---
> > > >
> > > > ### R-vp37.5d
> > > > > What is the exact role of "event data" and how is it integrated?
> > > >
> > > > **Response**
> > > > Event data from event cameras captures per-pixel intensity changes at microsecond resolution, making it ideal for deblurring fast motion. In GeMS-E, we use the **Event-based Double Integral (EDI)** model to reconstruct sharp images from event streams. These deblurred images are then fed into the GeMS pipeline for pose estimation and scene reconstruction, improving robustness when all input views are severely blurred.
> > > >
> > > > ---
> > > >
> > > > ### R-vp37.6
> > > > > The presentation of Ablation Table 2 could be improved by clearly denoting each column header.
> > > >
> > > > **Response**
> > > > Thank you for your feedback. I hope you are referring to clearer row headers in the ablation table 2 (now it is Table 4). We have now updated column headers for better readability; please refer to Section D.3 in the appendix.
> > > >
> > > > ---

---

> > > > > ### Author Response · Authors · 2025-06-26
> > > > > **To Reviewer vp37**
> > > > >
> > > > > ### R-vp37.7a
> > > > > > Why each component is necessary for this specific problem
> > > > >
> > > > > **Response**
> > > > > Each component in our framework is necessary because it directly addresses a specific challenge in 3D reconstruction from motion-blurred images. VGGSfM is essential since traditional SfM methods like COLMAP fail under severe blur, whereas VGGSfM can robustly estimate camera poses and sparse point clouds even from highly degraded inputs (See Table 2, 3 and Figure 7). The MCMC-based probabilistic scene modeling is crucial for adaptively densifying and stabilizing the reconstruction when the initial point cloud is sparse or noisy—a frequent occurrence in extreme blur scenarios (See Figure 8, 9). Joint optimization is required to refine both camera trajectories and scene geometry together, which is critical for accurately modeling the continuous motion that causes blur and for correcting errors that would otherwise persist if optimized independently. When event data is available, event-based deblurring (as in GeMS-E) becomes indispensable for recovering sharp images and further improving reconstruction quality in the most challenging cases, where all input views are severely blurred. Our ablation studies (Table 4) confirm that removing any of these components leads to significant drops in performance, demonstrating that each plays an indispensable role in solving this problem.
> > > > >
> > > > > ---
> > > > >
> > > > > ### R-vp37.7b
> > > > > > What the novelty is in their combination
> > > > >
> > > > > **Response**
> > > > > The novelty of our approach lies in how the initializations from VGGSfM are tightly integrated with 3DGS-MCMC, a probabilistic scene modeling, joint optimization, and event-based deblurring (in GeMS-E) to form a unified and robust pipeline for 3D reconstruction directly from extremely motion-blurred images. VGGSfM provides robust, blur-tolerant initialization of camera poses and point clouds, which is critical for handling cases where COLMAP fails. These initializations are then refined through probabilistic 3DGS-MCMC, which adaptively densifies and stabilizes the scene, and joint optimization, which simultaneously corrects both trajectory and geometry to accurately model the physics of blur. When event data is available, event-based deblurring further strengthens the pipeline by recovering sharp details in the most challenging scenarios. This close integration allows each component to compensate for the limitations of the others, resulting in a self-correcting system that achieves high-fidelity 3D reconstruction from blur-only inputs.
> > > > >
> > > > > ---
> > > > >
> > > > > ### R-vp37.7c
> > > > > > How the combination is greater than the sum of its parts
> > > > >
> > > > > **Response**
> > > > > The combination is greater than the sum of its parts because all components are integrated into a single end-to-end differentiable pipeline after VGGSfM initialization. This integration enables gradients to flow across all modules, so that errors in one component—such as pose inaccuracies or sparse geometry—can be corrected by others during optimization. Specifically, VGGSfM provides blur-tolerant initialization but may yield sparse or noisy point clouds under extreme blur; 3DGS-MCMC compensates through probabilistic sampling, adaptively densifying geometry and suppressing artifacts where heuristic methods fail; joint optimization continuously aligns scene structure and camera trajectories with observed data using physics-based constraints; and event-based deblurring, when available, rescues catastrophic cases by injecting sharp priors. This creates a mutual refinement loop, where VGGSfM’s initialization enables MCMC to handle sparse inputs, MCMC’s densification informs joint trajectory optimization, joint optimization corrects pose-scene misalignments, and events enhance initialization in extreme cases. Unlike isolated techniques or simple ensembles, this tightly coupled system achieves robust reconstruction from blur-only inputs—something impossible with fragmented approaches. The pipeline’s emergent capability stems from bidirectional error correction, where limitations in one module are resolved by downstream components.

---

### Review · Reviewer_sPXH · 2025-06-14

**Summary Of Contributions:**

The authors addresses the task of 3D scene reconstruction with extreme motion blurred images. The proposed algorithm eliminate the use of sharp images for pose estimation, instead, the authors use VGGSfm for initial point cloud estimation. Besides, they optimize camera motion trajectory and Gaussian parameters jointly. The authors also test their frame on even-based data and achieve state-of-the-art results. Extensive experiments on real-world and synthetic datasets demonstrate the effectiveness of the proposed work.

**Audience:**

Yes

**Claims And Evidence:**

Yes

**Requested Changes:**

See weaknesses.

**Strengths And Weaknesses:**

Strength:
1. The work tackles with a practical setting (reconstruction under motion blur)
2. The authors do experiments on both real-world data and synthetic event-based data.

Weakness:
1. The paper is not easy to follow. It is observed that the authors reconstruct sharp 3D scene with extreme motion blurred images, as shown in Fig 1. This is counter-intuitive and the authors didn't explain the rationale behind the such deblur algorithm in the main paper. Although the readers could find some clues in appendix, it could be better to clarity this in main paper breifly.
2. In Tab 1, the GeMS-E outperforms ExBluRF and BAD-Gaussians with sharp Image Supervision, which makes me curious. Could the authors explain the reason in detail?

---

> ### Author Response · Authors · 2025-06-26
> **To Reviewer sPXH**
>
> We deeply appreciate the reviewer’s time and effort in thoroughly evaluating our manuscript. Your constructive remarks and insightful questions have significantly contributed to refining both the presentation and technical depth of our work. We have thoughtfully considered each of your points and incorporated the necessary revisions into the manuscript. Our detailed, point-by-point responses are provided below for clarity and transparency.
>
> ### R-sPXH.1
> > The paper is not easy to follow. It is observed that the authors reconstruct sharp 3D scene with extreme motion blurred images, as shown in Fig 1. This is counter-intuitive and the authors didn't explain the rationale behind the such deblur algorithm in the main paper. Although the readers could find some clues in appendix, it could be better to clarity this in main paper breifly.
>
> **Response**
> Thank you for highlighting this concern. We agree that reconstructing sharp 3D scenes from extremely motion-blurred images can seem counter-intuitive, so we have clarified this rationale in the main paper.
>
> Our approach is based on a **physics-inspired blur formation model**: motion blur is the result of integrating sharp images along the camera’s trajectory during exposure. Our method renders multiple latent sharp images at different points along the estimated camera motion path (for example, using 15 virtual camera positions) and averages them to synthesize the observed blurred input, closely following the real blur formation process.
>
> During **joint optimization**, we refine both the underlying sharp 3D scene representation (the Gaussians) and the camera motion trajectory so that the rendered, averaged image matches the observed blur. This allows us to recover the **latent sharp scene**, even from severely blurred inputs.
>
> Now we include the **blur image formation model (Section 2.1.2)**, **camera motion trajectory modeling (Section 2.1.3)**, and **joint optimization (Section 2.1.5)** in the main paper along with a **schematic illustration of our method in Figure 2**, to make this process more intuitive.
>
> ---
>
> ### R-sPXH.2
> > In Tab 1, the GeMS-E outperforms ExBluRF and BAD-Gaussians with sharp Image Supervision, which makes me curious. Could the authors explain the reason in detail?
>
> **Response**
> GeMS-E achieves superior performance primarily due to its **highly accurate initialization** and **advanced motion modeling**.
>
> Event-based deblurring (EDI) produces **sharp-enough images** for VGGSfM to estimate camera poses that are **very close to ground truth (mean APE: 0.0862)**, which is crucial for reliable 3D reconstruction under extreme motion blur. Compared to BAD-Gaussians, GeMS-E provides **robust probabilistic refinement through 3DGS-MCMC**, which adaptively densifies geometry using noise-aware sampling and is more resilient to pose errors than the heuristic-based approach in BAD-Gaussians.
>
> Additionally, GeMS-E’s **joint optimization of Bézier trajectories and Gaussian parameters** ensures **physical consistency**, whereas BAD-Gaussians’ linear/spline motion approximation can introduce geometric inaccuracies, as presented in Figure 12.
>
> In contrast to ExBluRF, GeMS-E excels at modeling **complex, continuous camera motion**: while ExBluRF’s voxel-based representation struggles with extreme motion blur, **GeMS-E leverages a bundle-adjusted radiance field representation** enabling higher-quality reconstructions.
>
> This combination of **accurate initialization using VGGSfM from EDI motion-blurred images** and **3DGS-MCMC joint optimization with sophisticated motion modeling** enables GeMS-E to consistently outperform both ExBluRF and BAD-Gaussians, even when those methods have access to **sharp image supervision**. Now we add these details in **Section 3.2**.

---

### Review · Reviewer_Q5Pv · 2025-06-18

**Summary Of Contributions:**

This paper proposes GeMS, a system designed to reconstruct 3D Gaussian Splatting from images with extreme motion blur. The authors replace COLMAP with VGGSfM to obtain the initial camera poses and point cloud. During the optimization process of 3DGS, both camera poses and Gaussians are jointly optimized. When event camera data is available, it is first used for deblurring, which subsequently improves the reconstruction performance. Experimental results demonstrate that the proposed method performs well under conditions of extreme motion blur.

**Audience:**

Yes

**Claims And Evidence:**

No

**Requested Changes:**

As detailed in the weakness section, the authors should clearly explain the experimental setup of BAD-Gaussians, provide stronger justification for design choices, include deblurring performance comparisons with existing methods, and conduct a more comprehensive ablation study.

**Strengths And Weaknesses:**

Strengths:
- The authors address a practical and challenging problem: reconstructing scenes from blurred images.
- I agree with the authors’ observation that using clear images to estimate camera poses, as done in some prior work, is often unrealistic and may lead to biased evaluations.

Weaknesses:
- The paper lacks novelty. Although it claims to be COLMAP-free, it simply replaces COLMAP with VGGSfM, which serves the same function—initializing poses and point clouds—using a different method. Additionally, the use of event camera data for deblurring appears only loosely connected to the rest of the pipeline. Furthermore, the use of event cameras for deblurring is not closely related to other parts of this work, which makes it seem fragmented rather than a complete piece of work.

- From my understanding, BAD-Gaussians also performs SfM from blurred images using COLMAP (see Table 3 in the BAD-Gaussians paper). In this case, the performance of the proposed method is inferior to that of BAD-Gaussians.

- The paper contains several unsupported claims. For instance, the authors assert that VGGSfM outperforms COLMAP under extreme motion blur, yet they provide neither justification nor experimental evidence to support this claim. Similarly, the advantage of using MCMC-based Gaussian Splatting over standard Gaussian Splatting in motion-blur scenarios is not explained or validated.

- Although the paper includes a section titled "Baseline Methods and Evaluation Metrics," it fails to define or describe the evaluation metrics used.

- The evaluation is limited to novel-view synthesis of sharp images. Unlike prior work, such as BAD-Gaussians, this paper does not provide a comparison of deblurring performance on the input views.

- The ablation study is insufficient. In particular, there is no comparison with results obtained using COLMAP, which is essential to justify the choice of VGGSfM.


---
**Final recommendation**
Thanks to the authors for their detailed response. However, I would still recommend a weak reject for the following reasons:
- This paper may be less interesting to the TMLR's audience. This paper focuses on the engineering staff by replacing the COLMAP/3DGS of previous methods with existing other techniques: VGGSfM/3DGS-MCMC. However, the reason behind the replacement is still not well studied. If VGGSfM performs better than COLMAP on extreme motion blur scenes because it use better deep learning based matching  methods, what about replacing the original COLMAP feature matching method with better one? And are there other methods better than VGGSfM, such as [VGGT](https://vgg-t.github.io/). In addition, the study of failure cases is not well provided, and COLMAP still outperforms VGGSfM in many scenarios (e.g., large-scale scenarios), and the use of VGGSfM may bring other drawbacks.
- The paper still requires major revision. The claim of COLMAP-free is unnecessary. Saying COLMAP-free is more about not using SfM methods, instead of using different SfM methods.
- Methods such as BAD-Gaussians are agnostic to SfM methods, replacing the original COLMAP initialization with VGGSfM on BAD-Gaussians should also be a meaningful baseline.

---

> ### Author Response · Authors · 2025-06-26
> **To Reviewer Q5Pv**
>
> **Comment:**
> First of all, we sincerely thank the reviewer for taking the time to carefully read our manuscript and for providing thoughtful and constructive feedback. Your insights have been invaluable in helping us improve the clarity, rigor, and overall quality of our work. We have carefully addressed each point below and made the corresponding revisions to the manuscript. Our responses are organized point-by-point for clarity.
>
>
> ### R-Q5Pv.1
> > The paper lacks novelty. Although it claims to be COLMAP-free, it simply replaces COLMAP with VGGSfM, which serves the same function—initializing poses and point clouds—using a different method. Additionally, the use of event camera data for deblurring appears only loosely connected to the rest of the pipeline. Furthermore, the use of event cameras for deblurring is not closely related to other parts of this work, which makes it seem fragmented rather than a complete piece of work.
>
> **Response**
> Our work addresses a highly relevant and practical challenge in 3D vision: reconstructing accurate 3D scenes directly from extremely motion-blurred images. This is a critical problem in many real-world scenarios, such as high-speed motion or low-light environments, where capturing sharp images is not feasible. Previous methods typically depend on sharp-image supervision (impractical) or traditional pipelines like COLMAP, which are not effective under severe blur, please, see Table 2.
>
> The utility of our approach comes from tackling this open problem directly, using only blurred inputs. We achieve this through a carefully designed, integrated pipeline: **VGGSfM enables robust SfM initialization where COLMAP fails**, **3DGS-MCMC adaptively refines even sparse or noisy SfM initializations** via probabilistic modeling, and **joint optimization further improves both camera poses and scene geometry** in an end-to-end manner.
>
> Additionally, in the most challenging cases where all input views are severely blurred, if event data is available, **event-based deblurring is seamlessly incorporated** to provide sharp priors. These sharp deblurred images are directly passed through our GeMS framework in a plug-and-play approach. Each component is chosen and connected to address a specific limitation of existing methods, resulting in a unified solution that is both practical and effective for real-world, blur-dominated scenarios.
>
> ---
>
> ### R-Q5Pv.2
> > From my understanding, BAD-Gaussians also performs SfM from blurred images using COLMAP (see Table 3 in the BAD-Gaussians paper). In this case, the performance of the proposed method is inferior to that of BAD-Gaussians.
>
> **Response**
> While it is true that BAD-Gaussians uses COLMAP for SfM on blurred images, it is important to note that the BAD-Gaussians paper **does not address or evaluate performance under extreme motion blur**, focusing instead on **moderate blur scenarios where COLMAP can still function** (Table 3 in the BAD-Gaussians paper).
>
> In contrast, **our method is specifically designed for and extensively evaluated on extreme motion blur conditions**, where COLMAP-based approaches, including BAD-Gaussians, typically fail due to unreliable feature matching and pose estimation (see Table 2). By leveraging **VGGSfM for initialization**, which is robust to severe blur, and integrating probabilistic scene modeling and joint optimization, **our pipeline consistently achieves high-quality reconstructions** in settings where BAD-Gaussians is not applicable or effective.
>
> So, only for comparison purpose, we evaluated BAD-Gaussians with sharp supervision, as there is no other way in extreme motion blurred scenario. To address the reviewer's concern, we systematically evaluated both GeMS and BAD-Gaussians performance across increasing blur levels. COLMAP is able to provide valid SfM initializations only up to moderate blur (7-frame averaging); beyond this, it fails completely due to its reliance on feature matching, which is rendered ineffective by severe motion blur (see Table 2). In contrast, VGGSfM successfully produces robust initializations even at extreme blur levels (up to 11-frame averaging). Hence BAD-Gaussians works till moderate blur levels.
>
> To further compare downstream performance, we fed COLMAP initializations (for the blur levels where it succeeded) to BAD-Gaussians and VGGSfM initializations to our GeMS framework. As shown in Figure 15 and Table 11, **GeMS consistently outperforms BAD-Gaussians across all blur levels**, and as blur severity increases, the performance of BAD-Gaussians drops sharply while GeMS remains robust.
>
> ---

---

> > ### Author Response · Authors · 2025-06-26
> > **To Reviewer Q5Pv**
> >
> > ### R-Q5Pv.3
> > > The paper contains several unsupported claims. For instance, the authors assert that VGGSfM outperforms COLMAP under extreme motion blur, yet they provide neither justification nor experimental evidence to support this claim. Similarly, the advantage of using MCMC-based Gaussian Splatting over standard Gaussian Splatting in motion-blur scenarios is not explained or validated.
> >
> > **Response**
> > Thank you for your feedback. We now provide clear experimental evidence supporting our claims through systematic evaluations. First, we assessed VGGSfM and COLMAP across varied motion blur levels, **demonstrating VGGSfM's superior robustness under extreme blur where COLMAP fails** (Table 2). Also, **Ablations Table 4 (rows 6 and 5)** reports VGGSfM performance with respect to COLMAP — VGGSfM performs better.
> >
> > Second, we compared **3DGS-MCMC with standard 3DGS**, confirming that **probabilistic Gaussian Splatting significantly outperforms heuristic methods in motion-blur scenarios** (Figure 9). Third, by replacing input sharp point clouds with progressively blurred versions, we validated 3DGS-MCMC's robustness to blurry point cloud initializations: **it maintains consistent performance regardless of blur severity, while standard 3DGS degrades sharply** (Figure 8). Hence, it proves 3DGS-MCMC is robust to blurry point cloud initializations.
> >
> > These experiments collectively justify our pipeline's advantages in handling extreme motion blur challenges.
> >
> > ---
> >
> > ### R-Q5Pv.4
> > > Although the paper includes a section titled "Baseline Methods and Evaluation Metrics," it fails to define or describe the evaluation metrics used.
> >
> > **Response**
> > We apologize for the omission of metric definitions in the original submission. We have now added clear descriptions of all evaluation metrics used in our experiments, including **PSNR and SSIM for image quality assessment**, and **LPIPS for perceptual similarity**, which measures how close the generated images are to ground truth in a way that aligns with human perception.
> >
> > ---
> >
> > ### R-Q5Pv.5
> > > The evaluation is limited to novel-view synthesis of sharp images. Unlike prior work, such as BAD-Gaussians, this paper does not provide a comparison of deblurring performance on the input views.
> >
> > **Response**
> > Now we include comprehensive **deblurring metrics on input views** in our evaluation, which are provided in the appendix due to space limitations in the main paper (see Table 7). These results demonstrate our method's effectiveness for both **novel view synthesis** (see Table 1, Figure 14) and **direct deblurring of input frames** (see Table 7, Figures 4 & 5) under extreme motion blur.
> >
> > ---
> >
> > ### R-Q5Pv.6
> > > The ablation study is insufficient. In particular, there is no comparison with results obtained using COLMAP, which is essential to justify the choice of VGGSfM.
> >
> > **Response**
> > To address the reviewer's concern about COLMAP's limitations under extreme motion blur, we clarify that **COLMAP fails entirely in these scenarios** due to its reliance on feature matching, which becomes impossible when severe blur obscures image textures (see Table 2). Since COLMAP produces no usable output under extreme blur, direct ablation comparisons are infeasible — precisely why we propose the VGGSfM-based GeMS pipeline.
> >
> > To address the reviewer's concern regarding ablation with COLMAP, we systematically evaluated both COLMAP and VGGSfM across increasing blur levels. **COLMAP is able to provide valid SfM initializations only up to moderate blur (7-frame averaging); beyond this, it fails completely** due to its reliance on feature matching, which is rendered ineffective by severe motion blur (see Table 2). In contrast, **VGGSfM successfully produces robust initializations even at extreme blur levels (up to 11-frame averaging)**. This clearly demonstrates the necessity and effectiveness of VGGSfM in our pipeline, especially in scenarios where COLMAP fails.
> >
> > **Our ablations Table 4 (rows 6 and 5)** reports VGGSfM performance with respect to COLMAP — VGGSfM performs better.

---

> ### Author Response · Authors · 2025-07-21
> **Authors Response to Reviewer Q5Pv**
>
> We sincerely thank the reviewer for the feedback. Below, we respond to the three major concerns raised in the final recommendation.
>
> ---
>
> > *“This paper may be less interesting to the TMLR's audience. This paper focuses on the engineering staff by replacing the COLMAP/3DGS of previous methods with existing other techniques: VGGSfM/3DGS-MCMC. However, the reason behind the replacement is still not well studied. If VGGSfM performs better than COLMAP on extreme motion blur scenes because it use better deep learning based matching methods, what about replacing the original COLMAP feature matching method with better one? And are there other methods better than VGGSfM, such as VGGT. In addition, the study of failure cases is not well provided, and COLMAP still outperforms VGGSfM in many scenarios (e.g., large-scale scenarios), and the use of VGGSfM may bring other drawbacks.”*
>
> We would like to respectfully clarify why we believe our work is highly relevant to the TMLR audience. This paper contributes to an important and timely direction at the intersection of **learning-based 3D vision**, **rendering**, **trajectory optimization** and **physics-aware reconstruction**, particularly in the context of **3D Gaussian Splatting (3DGS)**, a technique that has rapidly gained traction in both academic and applied settings. While existing 3DGS pipelines typically assume **sharp or moderately blurred images** and **smooth camera motion**, our work addresses the significantly more challenging and underexplored setting of **extreme motion blur**, where standard pipelines fail. To address this, we propose a unified and differentiable framework that integrates **blur-tolerant SfM initialization**, **probabilistic scene modeling**, and **physically grounded joint optimization**, and extends to an **event-aware variant (GeMS-E)** that achieves robust reconstruction even in degenerate cases like where all input images are serverely blurred.
>
> We believe this contribution goes beyond component-level engineering: it offers a new direction for **blur-aware scene reconstruction** that is both **principled and practically robust**, grounded in the physics of image formation. The approach has direct implications for **robotics**, **autonomous systems**, and **perception under uncertainty** - all core domains of interest to the TMLR community.
>
> We have already addressed the rationale for replacing **COLMAP with VGGSfM** clearly in **Table 2, Table 3, and Figure 7**, which demonstrate VGGSfM’s superior robustness under extreme motion blur. Likewise, the motivation for replacing **standard 3DGS with 3DGS-MCMC** is supported by **Figure 8 and Figure 9**, which show that probabilistic refinement leads to significantly more stable and accurate reconstructions from blurry inputs.
>
> Regarding the suggestion to improve COLMAP’s front-end with better feature matchers:
> We explicitly evaluated this using **HLOC**, an SfM pipeline that integrates advanced feature matchers such as **SuperGlue**. However, as shown in **Table 4, Row 2**, the downstream reconstruction quality using HLOC under extreme motion blur is significantly lower. In contrast, **GeMS with VGGSfM (Table 4, Row 3)** achieves approximately **1.45 dB higher PSNR**, indicating that improved feature matching alone does not suffice - robust initialization under blur requires more than stronger correspondences.
>
>
> **Scope clarification**: While COLMAP or VGGT may outperform VGGSfM in large-scale scenarios, our work specifically targets **small-scale scenes with severe motion blur**, which pose distinct and underexplored challenges. Large-scale adaptation would require methods like VGGT and hierarchical Gaussian Splatting - which are orthogonal and out of scope for our current work.
>
> ---
>
> > *“The paper still requires major revision. The claim of COLMAP-free is unnecessary. Saying COLMAP-free is more about not using SfM methods, instead of using different SfM methods.”*
>
> We appreciate this feedback and agree that the phrase “COLMAP-free” may cause confusion.
>
> - In our original draft, this term appeared **twice in the introduction** and **twice in the method section**, to emphasize that we do not use COLMAP for SfM initialization - unlike most prior 3DGS pipelines.
> - However, to avoid any misinterpretation that we are SfM-free (which we are not), **we have now removed all instances of “COLMAP-free” from the manuscript**. This revision is purely terminological and **does not affect the substance or scope of our technical contributions**.
>
> We no longer claim that our method is COLMAP-free. Instead, we now describe our contribution as:
>
> > **“An efficient 3D Gaussian Splatting pipeline for handling extreme motion blur.”**
>
> This phrasing better reflects our technical scope and avoids ambiguity.
>
> ---

---

> > ### Author Response · Authors · 2025-07-21
> > **Authors Response to Reviewer Q5Pv**
> >
> > > *“Methods such as BAD-Gaussians are agnostic to SfM methods, replacing the original COLMAP initialization with VGGSfM on BAD-Gaussians should also be a meaningful baseline.”*
> >
> > We have already evaluated this variant in our ablation studies. Specifically:
> >
> > - In **Table 4**, **Row 1** reflects **BAD-Gaussians initialized with VGGSfM**, while **Row 3** corresponds to our **GeMS pipeline (also initialized with VGGSfM)**. GeMS achieves **0.47 dB higher PSNR** than BAD-Gaussians in this direct comparison, highlighting the benefits of probabilistic modeling (MCMC).
> >
> > - Similarly, **Rows 4 and 6** correspond to the **event-based deblurring setting**: **Row 4** is BAD-Gaussians initialized with VGGSfM using deblurred images from event-based initialization, and **Row 6** is our **GeMS-E variant** under the same input. GeMS-E achieves **0.89 dB higher PSNR**, demonstrating that even with event-based initialization, MCMC refinement leads to superior reconstruction quality.
> >
> > - Additionally, **Figure 12** shows that our **Bézier-based trajectory modeling** outperforms the **linear/spline motion representations** used in BAD-Gaussians, especially under extreme camera motion blur.
> >
> > These comparisons show that while BAD-Gaussians can be initialized with VGGSfM, it lacks **probabilistic scene modeling**, and uses a **simpler trajectory representation (linear/spline)** that is less expressive than our **Bézier-based modeling** - both of which are critical for robustness in extreme motion blur.
> >
> > Finally, our **GeMS-E variant**, which uses event-based deblurring only during initialization, **outperforms all BAD-Gaussians variants**, even those using sharp supervision - as shown in **Table 1, GeMS-E (Ours) column**. GeMS-E produces **artifact-free reconstructions** in severely motion-blurred scenarios.
> >
> > ---
> >
> > We respectfully believe these clarifications address the reviewer’s concerns, and we hope the paper’s contributions are now recognized more clearly.

---

### Decision · Action_Editor_p69t · 2025-07-20

**Recommendation:** Reject

**Additional Comments:**

N/A

**Audience:**

Yes

**Audience Explanation:**

The proposed work may be of interest to a small subset of the TMLLR community.

**Claims And Evidence:**

No

**Claims Explanation:**

The majority of the reviews lean toward rejecting this paper for several reasons.

First, this proposed method lack generalizable sufficient insights. More generalizable insights, such as a systematic analysis on how the deep learning methods affect the deblurring reconstruction, will be more interesting to the TMLR's audience.

Second, the claims of this paper are not always supported by convincing and clear justifications.
* Numerous methods such as BAD-Gaussians are agnostic to SfM methods; replacing the original COLMAP initialization with VGGSfM on BAD-Gaussians should also be a meaningful and necessary baseline for claiming the proposed pipeline is better compared with previous methods besides the changing of the SfM method.
* The claim of COLMAP-free is unnecessary and misleading. Saying COLMAP-free is more about not using SfM methods, instead of using different SfM methods.
* The study of failure cases is not well provided, and COLMAP still outperforms VGGSfM in many scenarios (e.g., large-scale scenarios), and the use of VGGSfM may bring other drawbacks. It also makes this work less convincing.

Third, it is difficult to precisely identify the value of the work due to the lack of clarity in both the method and writing.  Further refinement and clearer articulation of the contributions would help in better showing the significance and applicability of the proposed approach.

Finally, this paper may be less interesting to the TMLR's audience. This paper focuses on the engineering staff by replacing the COLMAP/3DGS of previous methods with existing other techniques: VGGSfM/3DGS-MCMC. However, the reason behind the replacement remains poorly understood. If VGGSfM performs better than COLMAP on extreme motion blur scenes because it uses better deep learning based matching methods, what about replacing the original COLMAP feature matching method with a better one? And are there other methods better than VGGSfM, such as VGGT. In addition, the study of failure cases is not well-supported, and COLMAP still outperforms VGGSfM in many scenarios (e.g., large-scale scenarios), suggesting that the use of VGGSfM may also bring other drawbacks.

Taking all factors into account, the editor does not recommend publication in TMLR.

---

> ### Author Response · Authors · 2025-07-31
> **Authors' Clarification Regarding Final Decision**
>
> We sincerely thank the Action Editor and reviewers for handling our submission, *“GeMS: Efficient Gaussian Splatting for Extreme Motion Blur.”*
>
> We would like to respectfully clarify that the concerns mentioned in the final decision closely resemble those raised earlier by Reviewer Q5Pv. We had already addressed these concerns thoroughly and in good faith within just a few days of their response. These responses are documented as the **3rd and 4th replies** under Reviewer Q5Pv’s thread on this page.
>
> Following those clarifications, we did not receive any further response from that reviewer. Additionally, none of the other reviewers responded at any point during the post-rebuttal phase, despite the expectation of active discussion in TMLR's review model. Given this, we had assumed that our clarifications were satisfactory and that the discussion phase had concluded.
>
> If there is any chance that those follow-up responses were overlooked during the final decision process, we would sincerely appreciate it if the Action Editor could take a moment to review them.
>
> We thank the community once again for the opportunity to participate in the TMLR review process and appreciate the time and feedback from all parties involved.

---

> > ### Comment · Editors_In_Chief · 2025-08-06
> >
> > Dear authors, here is some more information from the recommendation phase. Four out of the five reviewers said that the claims made in the paper were not met with suitable evidence. Two out of the five reviewers said that there wouldn't be an audience at TMLR for this paper.
> >
> > Here are their descriptive comments. While the paper cannot be accepted, hope that these reviews and comments will help improve the paper.
> >
> > ----
> >
> > Firstly, this work may be less interesting to the TMLR's audience.
> >
> > This paper focuses on the engineering staff by replacing the COLMAP/3DGS of previous methods with existing other techniques: VGGSfM/3DGS-MCMC. However, the reason behind the replacement remains poorly understood, making it difficult to draw generalizable insights. For instance, if VGGSfM performs better than COLMAP on extreme motion blur scenes because it uses better deep learning based matching methods, what about replacing the original COLMAP feature matching method with a better one? Are there other methods that are better than VGGSfM, such as VGGT? More generalizable insights, such as a systematic analysis on how the deep learning methods affect the deblurring reconstruction, will be more interesting to the TMLR's audience.
> > Secondly, the claims of this paper are not always supported by convincing and clear evidence.
> >
> > Methods such as BAD-Gaussians are agnostic to SfM methods; replacing the original COLMAP initialization with VGGSfM on BAD-Gaussians should also be a meaningful and necessary baseline for claiming the proposed pipeline is better compared with previous methods besides the changing of the SfM method.
> > The claim of COLMAP-free is unnecessary and misleading. Saying COLMAP-free is more about not using SfM methods, instead of using different SfM methods.
> > The study of failure cases is not well provided, and COLMAP still outperforms VGGSfM in many scenarios (e.g., large-scale scenarios), and the use of VGGSfM may bring other drawbacks. It also makes this work less conincing.
> >
> > ---
> >
> > Thank the authors for their response. I would recommend a leaning reject for the following reasons:
> >
> > The new novel view synthesis results in Figure 14 are confusing. In many cases, there are no obvious advancements of the proposed method, and the colored boxes notation is not well organized.
> > Now, Table 1 and Table 7 are duplicated. This and the above point weaken the accuracy and clearness of the evidence behind this paper's claims
> > I think this paper is less interesting to the audience of TMLR. It seems the authors are trying to show that VGGSfM is better than Colmap in scenarios where extremely blurry images are presented, but the paper is not presented in a way for making this claim. Meanwhile, the introduction of event camera makes the pipeline seem fragmented rather than a well-represented complete pipeline. I would recommend that the author reformulate the claims and show interesting findings and evidence in a clearer and organized way.
> >
> > ------
> >
> > Some of the concerns have been partially resolved in the rebuttal. However, in line with comments from other reviewers, I find that the paper still lacks clear and convincing evidence for each of its core components. Even after the rebuttal, the individual contributions come across as straightforward applications of existing techniques, and the proposed combination appears more like a simple aggregation than a well-justified, synergistic integration.
> >
> > Regarding the second criterion, I do not rule out potential interest from some members of the TMLR audience. However, due to the lack of clarity in both the method and writing, it is difficult to precisely identify the value of the work. Further refinement and clearer articulation of the contributions would help in better showing the significance and applicability of the proposed approach.
> >
> > -----
> >
> > The paper addresses the problem of images containing extreme motion blur. However, the key idea is just to adopt the VGGSfM to handle the pose problem, while no convincing motivation or explanation are provided. In my opinion, blurry images could have several plausible poses, while the proposed method does not handle this problem, which makes me feel less convinced about the proposed strategy. Thus, I tend to reject the paper.
> >
> >
> > -----
> >
> >
> > Theauthors handles with a practical task, reconstruction under extremenly blur images, and the proposed method performs better than previous state-of-the-art. Thus I give accept.